# Cocaine induces locomotor sensitization through a dopamine-dependent VTA-mPFC-FrA cortico-cortical pathway in male mice

Lun Wang [1,2,3,8], Min Gao [1,2,3,4,8], Qinglong Wang[1,2,3,8], Liyuan Sun[1,2,3], Muhammad Younus[1,2,3], Sixing Ma[1,2,3], Can Liu[1,2,3], Li Shi[1,2,3], Yang Lu[1,2,3], Bo Zhou[1,2,3], Suhua Sun[1,2,3], Guoqing Chen[1,2,3], Jie Li[1,2,3], Quanfeng Zhang[1,2,3], Feipeng Zhu[1,2,3,5] ✉, Changhe Wang [6,7] ✉ & Zhuan Zhou [1,2,3] ✉

As a central part of the mammalian brain, the prefrontal cortex (PFC) has been implicated in regulating cocaine-induced behaviors including compulsive seeking and reinstatement. Although dysfunction of the PFC has been reported in animal and human users with chronic cocaine abuse, less is known about how the PFC is involved in cocaine-induced behaviors. By using two-photon $Ca^{2+}$ imaging to simultaneously record tens of intact individual networking neurons in the frontal association cortex (FrA) in awake male mice, here we report that a systematic acute cocaine exposure decreased the FrA neural activity in mice, while the chemogenetic intervention blocked the cocaine-induced locomotor sensitization. The hypoactivity of FrA neurons was critically dependent on both dopamine transporters and dopamine transmission in the ventromedial PFC (vmPFC). Both dopamine D1R and D2R neurons in the vmPFC projected to and innervated FrA neurons, the manipulation of which changed the cocaine-induced hypoactivity of the FrA and locomotor sensitization. Together, this work demonstrates acute cocaine-induced hypoactivity of FrA neurons in awake mice, which defines a cortico-cortical projection bridging dopamine transmission and cocaine sensitization.

As one of the most abused drugs, cocaine acts by blocking the dopamine transporter (DAT) in the brain to become stimulating, reinforcing, and addictive[1–5]. The altered brain synaptic transmission modulated by the increased mesolimbic dopamine is one of the essential mechanisms underlying cocaine-induced place preference

and compulsive seeking behaviors, among which the dopaminergic projection from the ventral tegmental area (VTA) to the nucleus accumbens (NAc) has been well-established in mediating some of the cocaine-induced behaviors[1,6,7]. However, the knowledge of subcortical mesostriatal dopamine has not yet led to effective

[1]State Key Laboratory of Membrane Biology and Beijing Key Laboratory of Cardiometabolic Molecular Medicine, Institute of Molecular Medicine, College of Future Technology, Peking University, Beijing 100871, China. [2]Peking-Tsinghua Center for Life Sciences, Peking University, Beijing 100871, China. [3]PKU-IDG/McGovern Institute for Brain Research, Peking University, Beijing 100871, China. [4]Joint Graduate Program of Peking-Tsinghua-NIBS, Academy for Advanced Interdisciplinary Studies, Peking University, Beijing 100871, China. [5]State Key Laboratory of Molecular Developmental Biology, Institute of Genetics and Developmental Biology, Chinese Academy of Sciences, Beijing 100101, China. [6]Neuroscience Research Center, Institute of Mitochondrial Biology and Medicine, Key Laboratory of Biomedical Information Engineering of Ministry of Education, School of Life Science and Technology, Xi'an Jiaotong University, Xi'an 710049, China. [7]Department of Neurology, the First Affiliated Hospital of Xi'an Jiaotong University, Xi'an 710061, China. [8]These authors contributed equally: Lun Wang, Min Gao, Qinglong Wang. ✉e-mail: flysummer2006@126.com; changhecool@163.com; zzhou@pku.edu.cn

treatment of substance use disorders[8]. Thus, a further investigation of brain circuit mechanisms underlying substance use disorders is required.

The prefrontal cortex (PFC), especially the medial PFC (mPFC), has been reported to be abnormal in cocaine abusers, including human beings and rodents[9–11]. Anatomically, the PFC is subdivided into a number of subregions, including the frontal association cortex (FrA) and the mPFC, while the mPFC can be further dissociated into the anterior cingulate cortex, the prelimbic cortex (PrL), the infralimbic cortex (IL), and the dorsal peduncular cortex (DP). Some of the mPFC subregions, such as the PrL and IL, are functionally involved in the regulation of cocaine-seeking behaviors. Furthermore, optogenetic manipulations of mPFC neurons can enhance or impair the performance of compulsive cocaine-seeking in rodents[9,10,12–14]. The mPFC has also been reported, at the behavior level, to be related to cocaine-induced reinstatement and locomotor sensitization[15,16]. Although dopamine suppression of the mPFC has been proposed to mediate cocaine effect[17–21], the responses of mPFC neurons to VTA stimulations are manifold[22,23] and changes of a specific subtype of mPFC neurons remain largely unknown, especially in the awake mice with intact circuits of various spatial-temporal inputs, modulators, and different levels of network activity[24,25]. Mechanistically, D1 and D2 dopamine receptors are thought to be excitatory and inhibitory, respectively[24–28], however, there are also opposite reports that D1R is inhibitory and D2R is excitatory in a subset of cortical pyramidal neurons[29,30]. Thus, the specific responses of different subtypes of mPFC neurons depend on the firing pattern of dopaminergic terminals, local interaction with inhibitory interneurons, extracellular dopamine concentration, and D1R/D2R binding affinity[24,31]. These studies provide profound evidence for the involvement of the mPFC in cocaine-induced behaviors; however, both phenotypes and the underlying mechanisms remain elusive[24,25].

The FrA is located in the anterior dorsal lateral part of the PFC, receiving inputs from several regions like the agranular insular cortex, the anterior part of the basolateral amygdaloid nucleus, and the perirhinal cortex[32–34], and projecting to midbrain dopamine neurons (including the VTA and the substantia nigra pars compacta, SNc) in a greater number and density of neurons than all mPFC subregions[35]. These anatomical structures suggest that the FrA is a potentially important region involved in dopamine-related functions. Recent studies indeed have reported critical roles of the FrA in anxiety and fear memory;[32–34] however, whether and how this region is functionally involved in cocaine-related mechanisms remain unclear.

To address how and to what extent the multiple PFC subregions of an awake mouse are affected by acute cocaine, which targets neural signaling via DAT, norepinephrine transporter (NET), or serotonin transporter[36], a more thorough investigation of specific PFC subregions in cocaine exposure at the single-cell level is required. Especially, dopamine transmission modulates cortical neurons in a manifold yet mysterious manner according to various studies in slices, organotypic cultures, and anesthetized or freely-moving animals[22–24,37–39]. Decoding how cocaine affects PFC neurons from the very beginning with single-neuron recordings in the awake brain would shed new light on both the mesocortical signaling pathway and the initiating mechanisms of acute cocaine exposure that lead to subsequent neural adaptations and drug abuse disorders.

In the present work, by using two-photon Ca²⁺ imaging to visualize neuronal activity in the FrA in awake mice at the resolution of single-cell-recordings and (up to) single-action potentials[32,40,41], we found that acute cocaine induced substantial hypoactivity of the FrA and this change was necessary for cocaine-induced locomotor sensitization. Importantly, we further identified a vmPFC–FrA pathway that mediates the cocaine-induced hypoactivity and locomotor sensitization in response to the altered dopamine transmission, providing a cortico-cortical circuit mechanism of cocaine in awake mice.

## Results

### Acute cocaine induces hypoactivity in the FrA

To investigate whether acute cocaine administration affects PFC neurons, we used two-photon Ca²⁺ imaging to record individual neuronal activity in layers II/III of the FrA in head-fixed awake mice (Fig. 1a). The spontaneous somatic Ca²⁺ signals (SSCSs) were acquired with the Ca²⁺ indicator Cal-520 (Fig. 1b) to measure neuronal population activity at single-action-potential-sensitivity[32,40]. Compared to the control group (intraperitoneal, i.p. saline), the percentage of silent cells increased from 4.8 to 30.5% and the averaged SSCS frequency decreased by ~53% within 10–20 min after cocaine injection (10 mg/kg, Fig. 1c–f). In some of these mice, we made longer time-elapse recordings and found that the inhibitory effect of cocaine on the FrA was reversible within 40–50 min after cocaine injection (Fig. S1b), and the SSCS frequency in the control group was stable until at least 50 min after saline injection (Fig. S1a). To further determine whether the excitation change mainly occurs in excitatory or inhibitory neurons, we next applied Ca²⁺ imaging by virally expressing genetically-encoded GCaMP6s. Using adeno-associated virus (AAV) with CaMKIIα-GCaMP6s (Fig. 1g), we confirmed that acute cocaine administration inhibited the SSCS frequency of excitatory neurons by ~51% (Fig. 1h and S2a). However, when the inhibitory neurons were visualized by injecting the Cre-dependent GCaMP6s-expressing virus into the FrA of vGAT-Cre transgenic mice (Fig. 1i), we did not observe a significant change after cocaine injection (Fig. 1j and S2b). Thus, our results demonstrated that acute cocaine exposure had a pronounced hypoactive effect, specifically on excitatory (but not inhibitory) neurons in the FrA.

To test whether this acute cocaine-induced hypoactivity in the FrA is critical for cocaine-induced behaviors, we combined a chemogenetic activation approach with the behavioral tests of conditioned place preference (CPP) and locomotor sensitization, to investigate the rescue effect of FrA manipulation on reward association memory and possibly sensitization of incentive motivation[42–44]. CaMKIIα-hM3Dq-mCherry expressing virus was bilaterally injected into the FrA to express hM3Dq in excitatory neurons (Fig. 1k). Two-photon imaging (Fig. S3a–c) and whole-cell patch-clamp electrophysiology (Fig. S3d) confirmed that activation of hM3Dq with clozapine-N-oxide (CNO) is capable of activating FrA excitatory neurons in vivo and ex vivo. The results showed that chemogenetic activation of excitatory neurons successfully prevented cocaine-induced hypoactivity in the FrA of awake mice (Fig. S3c). Then, CPP was assessed following daily pairing with cocaine (10 mg/kg, i.p., saline for control groups) on days 1–3, during which locomotor sensitization was recorded (Fig. 1l). Chronic Ca²⁺ imaging confirmed that cocaine did inhibit FrA excitatory neurons on the 3 consecutive training days (Fig. S4); thus CNO was injected (i.p.) 30 min before each cocaine-pairing session to test the effect of FrA excitation on cocaine-induced sensitization. The CPP score in the chemogenetic activation group (hM3Dq virus-injected mice) was similar to that of control virus-injected mice (Fig. 1m and S5a), suggesting that disruption of the cocaine-induced hypoactivity in the FrA does not prevent cocaine-induced CPP, and implying that FrA excitatory neurons are irrelevant to the cocaine-induced CPP or aversion. Strikingly, control mice showed clear locomotor sensitization to repeated cocaine exposure, and this was completely abolished in AAV-hM3Dq mice (Fig. 1n). As a control, both the basal locomotor activity (saline groups) and the cocaine-induced hyperactivity in day 1 remained unchanged in the hM3Dq group (Fig. 1n). Moreover, we applied the same behavioral assays to intact C57BL/6 J mice and excluded a possible role of CNO per se in regulating locomotor sensitization (Fig. S5b, c).

On the other hand, we tested whether inhibitory DREADDs (hM4Di) promote subthreshold cocaine-induced locomotor sensitization. We first examined the cocaine dose-dependent behaviors and selected 2.5 mg/kg as the subthreshold dose (Fig. S6a), which is only able to induce significant locomotor distance sensitization on day 3

but not on day 2. Then we followed the same protocol but used hM4Di instead of hM3Dq to investigate the role of FrA inhibition in the cocaine effect. The data show that locomotor sensitization of hM4Di-coc mice were indeed facilitated compared with the control group mCherry-coc (Fig. S6b2), leaving the CCP score unchanged (Fig. S6b1). The locomotor distance of hM4Di-coc mice was significantly increased on day 1 to a similar level as day 3 and thus didn't express further sensitization across the 3 days examination (Fig. S6b3). Thus,

chemogenetic inhibition of FrA neurons produced enhanced locomotor hyperactivity and thus, no further facilitation was observed. Collectively, these results demonstrated that acute cocaine-induced hypoactivity of the FrA is necessary for locomotor sensitization.

**Cocaine-induced hypoactivity of the FrA is dependent on DAT**

Although the dopamine system has been proposed to be the initiating target of cocaine to affect locomotion, reward, and compulsive

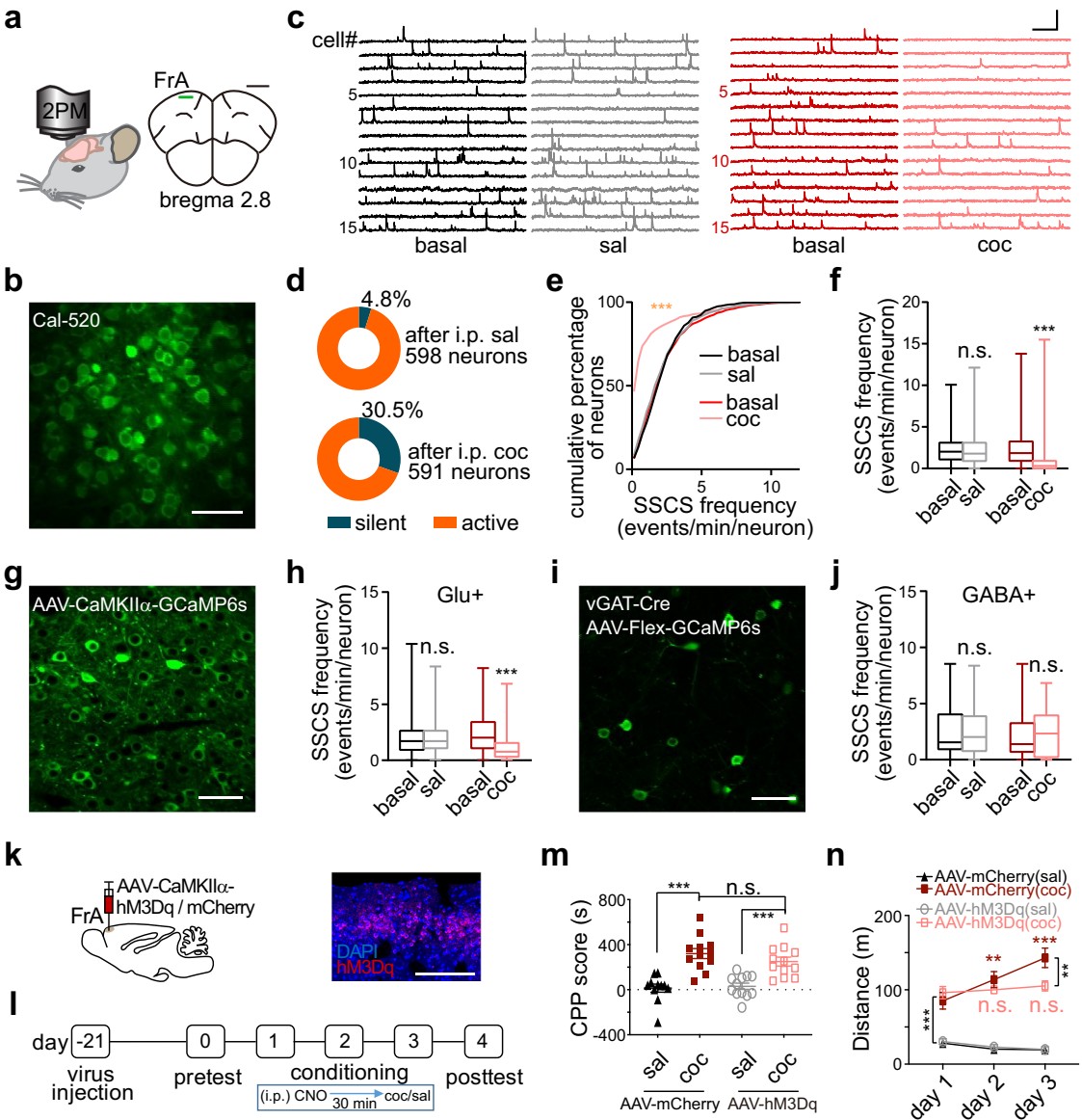

**Fig. 1 | Acute cocaine-induced hypoactivity of the FrA is necessary for locomotor sensitization. a, b** Schematic and representative micrograph showing two-photon Ca²⁺ imaging of Cal-520-labeled neurons in the FrA of head-fixed awake mice. Scale bars: 1 mm (**a**) and 50 μm (**b**). **c** Representative traces of spontaneous somatic Ca²⁺ signals (SSCS) before (basal) and after i.p. injection of saline (sal) or cocaine (coc, 10 mg/kg). Scale bars, 20 s, 200% dF/F₀. **d** Percentage of silent versus active cells after saline or cocaine injection. **e, f** Cumulative distribution and box-and-whisker plot (minimum, maximum, and three quartiles) of SSCS frequency under different conditions. Sal, $n = 598$ neurons from seven mice ($p = 0.28$); coc, $n = 591$ neurons from seven mice. **g, h** Representative micrograph and statistics (minimum, maximum, and three quartiles) of SSCS frequency of excitatory neurons in the FrA before and after i.p. saline or cocaine. Scale bar, 50 μm. Sal, $n = 185$ neurons from four mice ($p = 0.63$); coc, $n = 227$ neurons from five mice. **i, j** Representative micrograph and statistics (minimum, maximum, and three

quartiles) of SSCS frequency of GABAergic neurons in the FrA before and after saline or cocaine i.p. injection. Scale bar, 50 μm. Sal, $n = 35$ neurons from four mice ($p = 0.16$); coc, $n = 37$ neurons from four mice ($p = 0.21$). **k** Schematic of virus injection and representative micrograph showing the expression of hM3Dq-mCherry in excitatory FrA neurons; scale bar, 200 μm. **l** Timeline of behavioral tests. **m** Group data (mean ± s.e.m.) for CPP score. $n = 11$ (AAV-mCherry, sal), 12, 12, 12 mice. **n** Statistics (mean ± s.e.m.) of distance traveled for locomotor sensitization measurement. $n = 11$ (AAV-mCherry, sal), 12, 12, 12 mice. Two-tailed Wilcoxon test for (**e–j**), Ordinary two-way ANOVA followed by Bonferroni's multiple comparisons for (**m** and between-group comparisons in **n**), One-way ANOVA followed by Dunnett's multiple comparisons for (in-group comparisons in **n**). P values of (**m**, **n**) were summarized in Supplementary Table 1. n.s. not significant; *$p < 0.05$; **$p < 0.01$; ***$p < 0.001$. Source data are provided as a Source Data file.

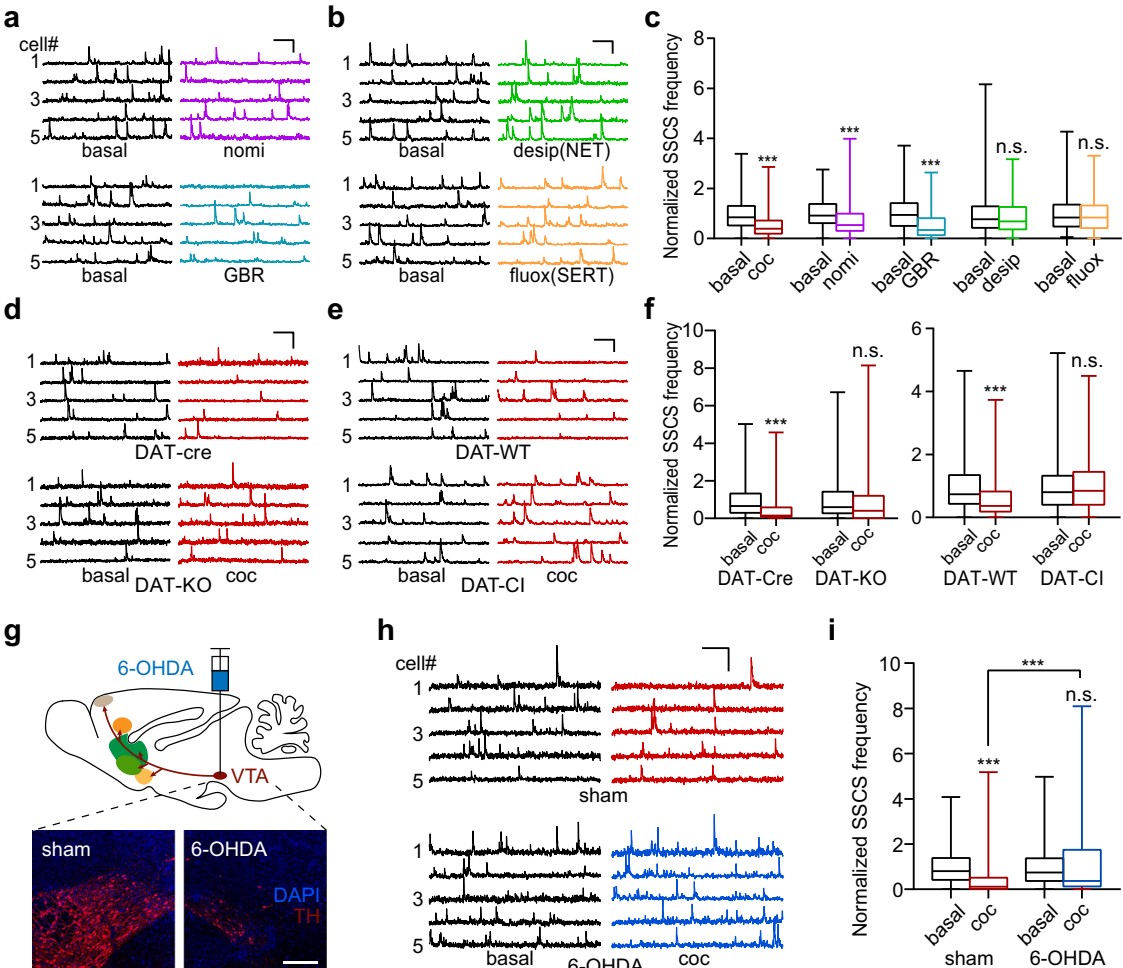

**Fig. 2 | Dopamine transporter (DAT) mediates cocaine-induced hypoactivity of the FrA. a–c** Representative traces and normalized frequency (minimum, maximum, and three quartiles) of SSCS of excitatory neurons in the FrA after i.p. cocaine, nomifensine (nomi, 8 mg/kg, a blocker of DAT and NET), GBR-12783 (GBR, 10 mg/kg, selective DAT blocker), desipramine (desip, 35 mg/kg, selective NET blocker), or fluoxetine (fluox, 10 mg/kg, selective SERT blocker). AAV-CaMKIIα-GCaMP6s was used to label excitatory neurons in FrA layers II/III. coc, $n = 285$ neurons from five mice; nomi, $n = 145$ neurons from three mice; GBR, $n = 201$ neurons from four mice; desip, $n = 208$ neurons from four mice ($p = 0.064$); fluox, $n = 281$ neurons from five mice ($p = 0.23$). **d–f** Representative traces and normalized frequency (minimum, maximum, and three quartiles) of SSCS in the FrA after i.p. cocaine. The Ca²⁺ indicator Cal-520 was used for Ca²⁺ imaging in DAT-KO and DAT-Cre mice. AAV-CaMKIIα-GCaMP6s was used in DAT-CI and DAT-WT mice. DAT-Cre,

$n = 173$ neurons from four mice; DAT-KO, $n = 202$ neurons from five mice ($p = 0.12$); DAT-WT, $n = 181$ neurons from four mice; DAT-CI, $n = 282$ neurons from five mice ($p = 0.28$). **g** Confirmation by TH-immunofluorescence (red) that VTA dopamine neurons are lesioned by stereotaxic microinjection of 6-OHDA. Representative micrographs are from the same region of sham- and 6-OHDA-treated mice and aligned symmetrically by the midline, $n = 4$, 4 mice. Scale bar, 200 μm. Cal-520 was used for imaging in the FrA. **h, i** Representative traces and normalized SSCS frequency (minimum, maximum, and three quartiles) in FrA neurons of sham- and 6-OHDA-treated mice before and after i.p. cocaine. Sham, $n = 151$ neurons from four mice; 6-OHDA, $n = 171$ neurons from four mice ($p = 0.27$). Two-tailed Wilcoxon test for (**c**, **f**, and paired comparisons in **i**) and two-tailed Mann–Whitney test for (unpaired comparisons in **i**). ***$p < 0.001$. Scale bars for traces, 20 s, 200% dF/F₀. Source data are provided as a Source Data file.

seeking[1–5], it is necessary, without bias, to identify the underlying target(s) of cocaine with regard to the FrA, which is a newly-identified area involved in cocaine action. As a nonselective inhibitor of monoamine transporters, cocaine inhibits all three kinds of monoamine transporter to a similar extent, including DAT, SERT, and NET[36,45]. We first injected (i.p.) relatively selective inhibitors of different transporters to determine which type of monoamine transporter is functionally involved in the cocaine-induced hypoactivity of the FrA. Interestingly, both nomifensine[32] and GBR-12783[46], specific DAT blockers, showed robust inhibition of FrA excitatory neurons, while desipramine (NET blocker)[47] and fluoxetine (SERT blocker)[48] did not (Fig. 2a–c), implying functional involvement of the dopamine system in the modulation of FrA activity. Importantly, we tested the acute cocaine effect on DAT knockout (DAT-KO)[49] and cocaine-insensitive DAT (DAT-CI)[3] transgenic mice and found that cocaine failed to inhibit FrA activity in either transgenic line (Fig. 2d–f), indicating critical roles of DAT in

cocaine-induced FrA hypoactivity. In addition, the inhibitory effect of cocaine on FrA neurons disappeared when VTA dopaminergic neurons were depleted by the stereotactic injection of the neurotoxin 6-hydroxydopamine (6-OHDA, Fig. 2g–i). In contrast, mice treated with MPTP (1-methyl-4-phenyl-1,2,3,6-tetrahydropyridine, i.p. daily for 5 days) or 6-OHDA (injected into SNc) showed substantial dopamine loss in the SNc and the caudate-putamen (CPu, dorsal striatum) while the cocaine-induced FrA hypoactivity remained unchanged (Fig. S7). Together, these results suggested that VTA dopaminergic neurons play essential roles in cocaine-induced hypoactivity in the FrA.

## Cocaine-induced FrA hypoactivity is not dependent on the direct dopaminergic projection

To further investigate whether the direct dopaminergic projection contributes to cocaine's effect on the FrA, we used local microinjection of 6-OHDA to deplete dopaminergic terminals in this region. Tyrosine

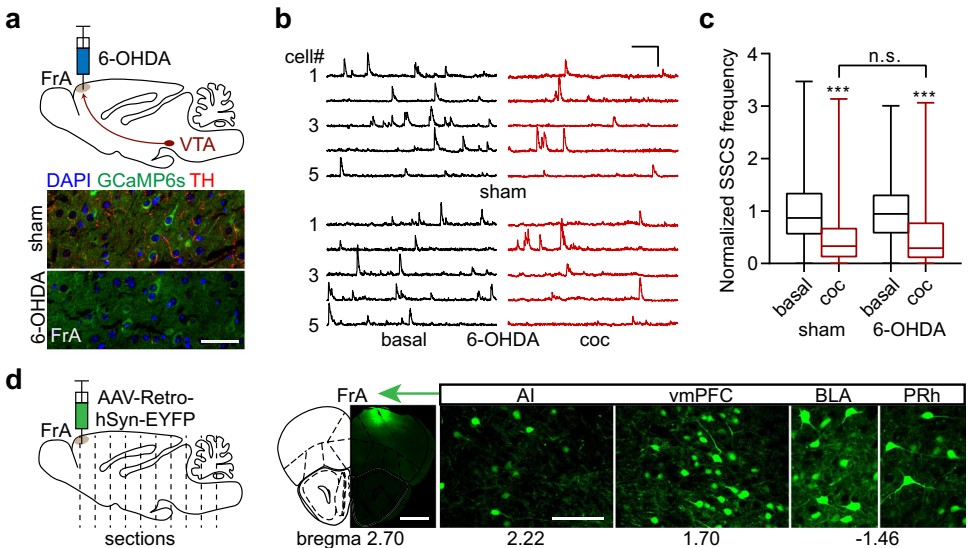

**Fig. 3 | Dopaminergic projection to the FrA is not necessary for cocaine-induced hypoactivity of FrA neurons. a** Upper, schematic of 6-OHDA injection into the FrA. Lower, TH-staining (red) showing the depletion of dopaminergic terminals in the FrA through 6-OHDA microinjection. Pyramidal neurons in the FrA are labeled with AAV-CaMKIIα-GCaMP6s. Scale bar, 50 μm. **b, c** Representative traces and normalized SSCS frequency (minimum, maximum, and three quartiles) of FrA pyramidal neurons in 6-OHDA-lesioned mice. Scale bars, 20 s, 300% dF/F₀. Sham, $n = 301$ neurons from 6 mice; 6-OHDA, $n = 218$ neurons from five mice,

$p = 0.35$ for coc (sham vs 6-OHDA), two-tailed Wilcoxon test for paired and Mann−Whitney test for unpaired comparisons, ***$p < 0.001$. **d** Left, schematic of AAV-Retro-hSyn-EYFP injection into the FrA for the retrograde labeling of FrA-projecting neurons. Right, representative micrographs of four regions with retrogradely-labeled somata (scale bars, 1 mm for FrA, 100 μm for projecting regions). AI agranular insular cortex, vmPFC ventromedial prefrontal cortex, BLA basolateral amygdaloid nucleus, anterior part, PRh perirhinal cortex. $n = 5$ mice. Source data are provided as a Source Data file.

hydroxylase (TH) staining (Fig. 3a and S8a, b) showed that the depletion was restricted to the FrA without obvious damage to other dopamine-projecting regions, including the vmPFC, NAc, CPu, or basolateral amygdaloid nucleus, anterior part (BLA) (Fig. S8c). Surprisingly, we found that cocaine still evoked a pronounced inhibition in the FrA similar to the sham group (Fig. 3a−c), indicating that the direct dopaminergic projection is not responsible for the inhibitory effect of cocaine on FrA activity.

To investigate which dopaminergic projection is responsible for the acute cocaine-induced hypoactivity in the FrA, we next screened the areas upstream of the FrA, especially those with well-known dopaminergic innervation, taking a retrograde tracing approach. The hSyn-EYFP retrograde AAV was injected into the FrA and images of consecutive sections of the whole brain showed densely-localized YFP-positive somata in several upstream areas, including the agranular insular cortex, BLA, and perirhinal cortex (Fig. 3d), consistent with previous reports[33,34]. Besides, we also found the vmPFC, including the IL and DP, to be positive with a relatively dense expression of YFP (Fig. 3d and S9), which had not been reported. Among them, the BLA and mPFC have been reported to be regulated by dopamine[23,50].

## Dopaminergic projections to the vmPFC mediate the effect of cocaine on the FrA

Next, we used a similar approach to investigate whether the BLA or vmPFC dopamine projection is necessary for the inhibitory effect of cocaine on the FrA. However, bilateral injection of 6-OHDA into the BLA (Fig. S10a) had no effect on the cocaine-induced FrA hypoactivity (Fig. S10b, c). In addition, in vivo two-photon imaging showed that cannula-guided micro-infusion of cocaine into the BLA failed to modulate neural activity in the FrA either (Fig. S10d, e). Thus, the BLA dopaminergic projection may be irrelevant to the effect of cocaine on the FrA.

On the contrary, the depletion of dopaminergic terminals in the vmPFC by microinjection of 6-OHDA notably blocked the cocaine-induced reduction in SSCS frequency (Fig. 4a−c). TH-staining showed

no obvious damage to dopaminergic terminals in other regions, including the FrA, NAc, CPu, or BLA (Fig. S11). Consistently, in vivo electrochemical amperometry recording with carbon fiber electrodes (Fig. 4d) showed that electrical stimulation-induced dopamine overflow in the vmPFC was increased in amplitude and decreased in uptake rate after cocaine administration (Fig. 4e, f). Importantly, micro-infusion of cocaine or dopamine into the vmPFC also induced hypoactivity of FrA neurons to an extent similar to that of i.p. cocaine injection (Fig. 4g−i). These results suggested that the vmPFC mediates the inhibition of FrA activity by cocaine.

## vmPFC−D1R and −D2R neurons mediate cocaine-induced hypoactivity of the FrA and locomotor sensitization

To investigate the vmPFC−FrA projections, the anterograde trans-monosynaptic tracing virus (AAV1-hSyn-Cre) was injected into the vmPFC to identify the postsynaptic neurons in the FrA (Fig. 5a−e). We found the majority of labeled neurons located in layers II/III of the FrA (mainly at a depth of 100−200 μm) and there was no anti-GAD67 fluorescence staining in these cells, suggesting a limited number of postsynaptic GABAergic neurons (Fig. 5d). Then, a retrograde AAV virus expressing YFP driven by CaMKIIα (AAV-Retro-CaMKIIα-EYFP) or vGAT (AAV-Retro-vGAT-EYFP) promotor was injected into the FrA of D1/D2-Cre and Ai9-crossed mice, in which the D1R- or D2R-positive neurons were labeled with tdTomato (Fig. 5f and S12a, b). We found that ~28% of the FrA-projecting excitatory vmPFC neurons and ~17% of the FrA-projecting inhibitory vmPFC neurons were D1R-positive, and ~25% excitatory and ~30% inhibitory neurons were D2R-positive (Fig. 5g and S12c). Averagely, among those FrA-projecting D1R and D2R neurons, about 81% were excitatory and 19% inhibitory for D1R, 68% excitatory, and 32% inhibitory for D2R (Fig. S12d).

Next, we combined optogenetic stimulation with a patch-clamp recording of synaptic currents to identify the functional synaptic connections between the vmPFC and the FrA (Fig. 5h). After the ChR2-expressing virus was injected into the vmPFC of D1- or D2-Cre mice, light-pulse stimulation (473 nm, 1 ms) not only evoked excitatory but

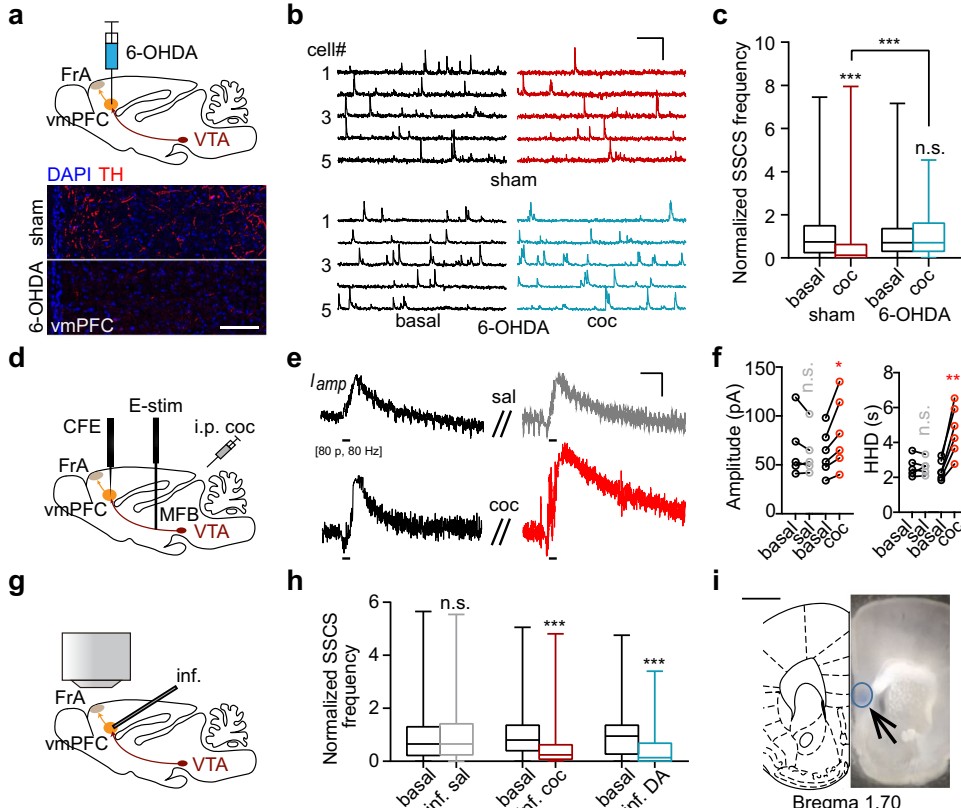

**Fig. 4 | Dopaminergic projection to the vmPFC is necessary for cocaine-induced hypoactivity of the FrA. a** Upper, schematic of 6-OHDA injection into the vmPFC. Lower, TH-staining (red) showing the depletion of dopaminergic terminals in the vmPFC through 6-OHDA microinjection. Cal-520 was used for $Ca^{2+}$ imaging in the FrA. Scale bar, 100 μm. **b, c** Representative traces (scale bars: 20 s, 200% $dF/F_0$) and normalized SSCS frequency (minimum, maximum, and three quartiles) of FrA neurons as in (**a**). Sham, $n = 406$ neurons from seven mice; 6-OHDA, $n = 397$ neurons from seven mice ($p = 0.77$), two-tailed Wilcoxon test for paired and two-tailed Mann–Whitney test for unpaired comparison. **d** In vivo amperometric CFE (carbon fiber electrode) recording of dopamine (DA) release in the vmPFC in response to electrical stimulation (E-stim, 1 ms, 80 pulses at 80 Hz) in the MFB (medial forebrain bundle). **e, f** Representative amperometric traces ($I_{amp}$, scale bars: 2 s, 30 pA) and statistics (mean ± s.e.m.) of vmPFC DA overflow amplitude and half height duration (HHD) in response to electrical stimulation applied before and after intraperitoneal injection of saline or cocaine (10 mg/kg). Saline, $n = 5$ mice ($p = 0.25$ for amplitude, 0.88 for HHD); cocaine, $n = 6$ mice ($p = 0.017$ for amplitude, 0.0019 for HHD), paired two-tailed Student's $t$-test. **g, h** Schematic of two-photon $Ca^{2+}$ imaging and statistics of normalized SSCS frequency (minimum, maximum, and three quartiles) of the FrA in response to microinfusion (inf.). Sal, $n = 191$ neurons from four mice ($p = 0.59$); coc, $n = 204$ neurons from four mice; DA, $n = 127$ neurons from three mice, two-tailed Wilcoxon test. **i** Confirmation of cannula tip location and drug diffusion through the microinfusion of Chicago sky blue. Scale bar, 1 mm. *$p < 0.05$, **$p < 0.01$, ***$p < 0.001$. Source data are provided as a Source Data file.

also inhibitory postsynaptic currents (EPSCs/IPSCs) in layers II/III of the FrA (Fig. 5i, j), which was consistent with the results of retrograde tracing assays (Fig. 5g). Notably, over half (55%) of the recorded FrA layer II/III neurons received vmPFC–D1R innervation (33% for D2), and ~50% (71% for D2) of those positive neurons received both EPSC and IPSC inputs, ~20% (21% for D2) EPSCs and ~30% (7% for D2) IPSCs (Fig. 5k). These structural and functional evidence demonstrated that the synaptic innervation from the vmPFC to the FrA is manifold and related to D1R-/D2R-expressing (vmPFC$^{D1R/D2R}$) neurons.

To investigate the roles of vmPFC$^{D1R/D2R}$ neurons in the cocaine-induced hypoactivity of the FrA in awake mice, we specifically manipulated the vmPFC$^{D1R/D2R}$ neurons by chemogenetic tools (Fig. 6a) and by micro-infusion of the selective D1R/D2R agonist or antagonist into the vmPFC (Fig. 6c). The results of single-neuron SSCSs showed that both excitatory manipulations of vmPFC$^{D1R}$ and vmPFC$^{D2R}$ neurons by hM3Dq produced significant depression of the FrA, by ~37 and ~19%, respectively (Fig. 6b). As a contrast, the inhibitory hM4Di failed to produce significant perturbation to the FrA activity in both D1 and D2 neurons (Fig. 6b). We also found that the cocaine-induced hypoactivity of the FrA was enhanced by SKF (SKF-38393, D1R agonist) and QP (quinpirole, D2R agonist), abolished by SCH (SCH-23390, D1R antagonist), and changed by sulpiride (D2R antagonist; Fig. 6d). These

findings demonstrate that D1 and D2 receptors play essential and different roles by bridging the dopamine transmission from the VTA–vmPFC projection and the vmPFC innervation of FrA excitatory neurons. Furthermore, both the antagonists of D1R and D2R micro-infused into the vmPFC were capable of blocking cocaine-induced locomotor sensitization (Fig. 6e). These results demonstrate a cortico-cortical pathway by which cocaine affects the FrA and locomotor sensitization in a dopamine- and D1/D2 receptor-dependent manner.

## Discussion

Previous studies have reported the prefrontal neuronal plastic changes induced by chronic cocaine use[1,9–12], but little is known about how the PFC responds to acute cocaine exposure and what the response means in awake mice. In the present work, we have identified the FrA, which is inhibited by acute cocaine exposure through the vmPFC–FrA pathway in a dopamine-dependent manner, as a critical mediator of locomotor sensitization (Fig. 7), providing a cortico-cortical pathway mediating cocaine sensitization in awake mice.

The first major finding of our study is that the FrA hypoactivity is responsible for cocaine-induced locomotor sensitization. Although the decreased activity or dysfunction of the PFC (hypofrontality) has been reported in animal[10,51] and human[11,52] abusers after chronic

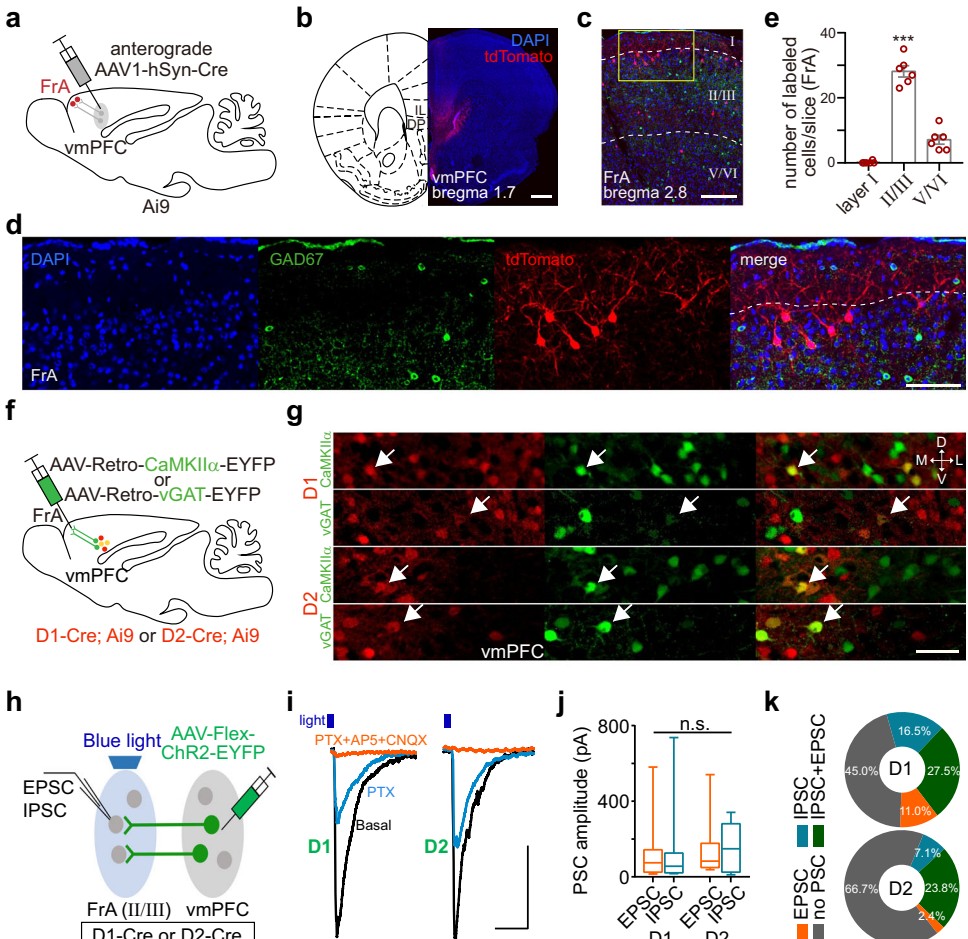

**Fig. 5 | vmPFC–D1R and -D2R neurons synapse to the FrA. a** Schematic of anterograde trans-monosynaptic virus (AAV1-hSyn-Cre) injection into the vmPFC (IL and DP) of Ai9 mice with Cre-dependent tdTomato expression. **b** Representative micrograph showing tdTomato expression in the vmPFC, repeated in three mice. Scale bar, 500 μm. **c, d** Representative micrograph of anterogradely-labeled neurons (red) in the FrA with immunofluorescence for GAD67 (green). The boxed region is enlarged in (**d**). Scale bars, 200 μm for (**c**), 100 μm for (**d**). **e** Statistics (mean ± s.e.m.) of anterogradely-labeled neurons in different layers of the FrA. $n = 6$ slices from three mice. Comparisons between layers II/III and layers I or V/VI were analyzed with one-way ANOVA followed by Dunnett's multiple comparisons: $F(1.606, 8.032) = 201.2$, $p < 0.001$; multiple comparisons: $p < 0.0001$. **f, g** Schematic and representative micrographs showing retrograde virus injection into the FrA of D1/D2-Cre and Ai9-crossed mice. White arrows indicate the retrograde virus-labeled D1R- or D2R-positive (red) glutamatergic or GABAergic neurons in the

vmPFC (IL and DP, coronal slices). Scale bar, 50 μm. **h** Schematic showing the recording of light-evoked EPSCs and IPSCs in layers II/III of acute FrA slices. ChR2 is expressed through stereotaxic injection of AAV-Flex-ChR2-EYFP into the vmPFC of D1-Cre or D2-Cre mice. **i, j** Representative traces and statistics (minimum, maximum, and three quartiles) of light-evoked PSCs ($n = 14, 14, 11, 12$ cells from 5 D1-Cre mice and 3 D2-Cre mice) in the FrA. Scale bars, 50 ms, 100 pA. Two-way ANOVA followed by Tukey's multiple comparisons test: D1/D2 effect, $F(1, 47) = 0.69$, $p = 0.41$; EPSC/IPSC effect, $F(1, 47) = 0.006$, $p = 0.94$; multiple comparison tests were all n.s. with $p > 0.9$. **k** Percentages of FrA neurons with IPSCs, EPSCs, or both. D1, $n = 60$ cells from six mice (33 PSC$^+$ cells, 27 PSC$^-$ cells); D2, $n = 75$ cells from seven mice (25 PSC$^+$ cells, 50 PSC$^-$ cells). Only 20 of the 33 PSC$^+$ cells and 14 of the 25 PSC$^+$ cells were tested for both EPSCs and IPSCs. Source data are provided as a Source Data file.

cocaine use or acute administration[21,53], what specific subregion is responsible for the cocaine effect remains obscure. With single-neuron-resolution of two-photon Ca$^{2+}$ imaging in awake mice, we show that most FrA neurons were reversibly inhibited by acute cocaine exposure (i.p.) and generated overall hypoactivity at the population level (Fig. 1 and S1). This inhibition was recorded only in excitatory but not inhibitory neurons in the FrA (Fig. 1g–j and S2). By combining two-photon imaging with pharmacological (antagonists of NET, SERT, and DAT) and genetic (DAT-KO and DAT-CI mice) tools, we further identified DAT as the major mediator of cocaine-induced hypoactivity in the FrA (Fig. 2). This is consistent with that most of the important psychological effects of cocaine are related to DAT and the dopamine system[1–3,5,6,22]. With chemogenetic manipulation, the FrA showed specificity in regulating sensitization without affecting the normal locomotion or rewarding effect (Fig. 1k–n and S3–S5). These findings not only suggest the close association of the FrA with locomotor

sensitization, a measurement associated with drug-induced plasticity and maybe incentive motivation[42–44], but also define this cortical region as a therapeutic target to specifically reverse cocaine sensitization. Because locomotor sensitization is associated with cocaine-induced neural plasticity and craving, which may lead to dependence and loss of self-control[42,44], future work is deserved to examine possible roles of the FrA using a behavior paradigm like self-administration in abstinence, compulsive drug-seeking, and relapse, to achieve a comprehensive understanding of the FrA in cocaine-abuse disorders.

The second finding is that the dopamine dependence was not through the direct dopamine innervation to the FrA. Instead, dopamine indirectly modulates the FrA activity through VTA projections to the vmPFC that synapses to FrA layer II/III neurons (Figs. 3–5 and S8–S12). As a control, dopaminergic projections to the BLA were not involved in the effect of cocaine on the FrA (Fig. S10), implying relative specificity of the VTA–vmPFC–FrA circuit in mediating

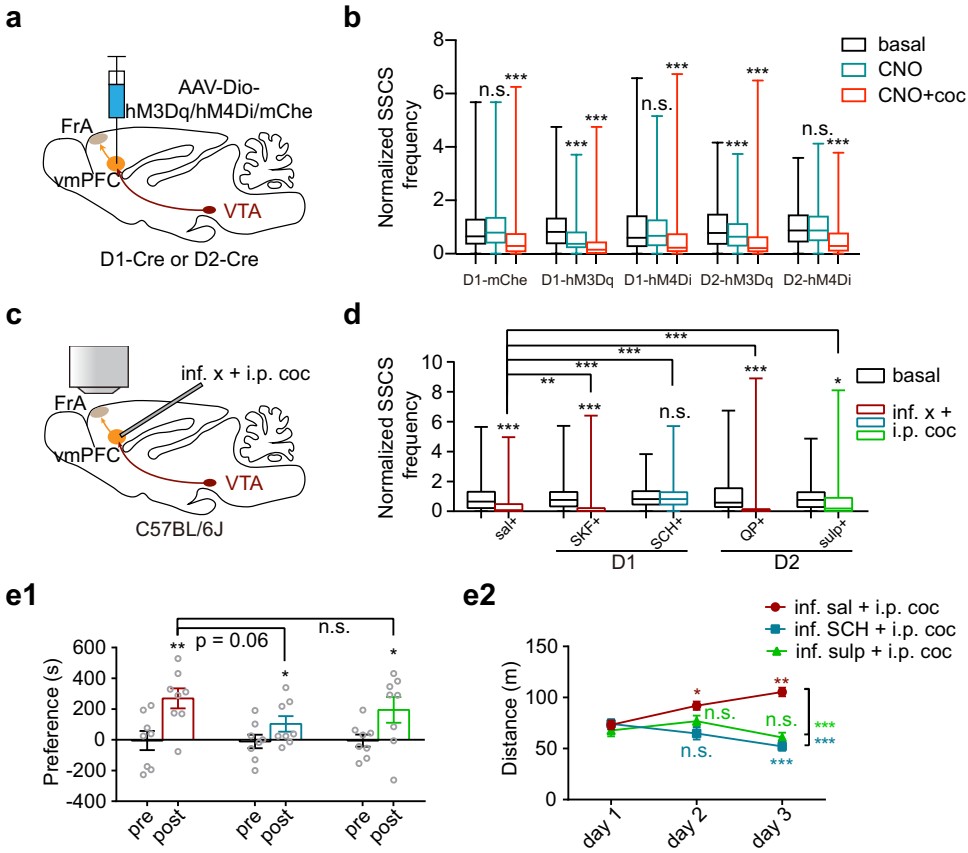

**Fig. 6 | vmPFC modulates FrA and locomotor sensitization through dopamine D1 and D2 receptors. a, b** Schematic of two-photon Ca²⁺ imaging and statistics (minimum, maximum, and three quartiles) of normalized SSCS frequency in the FrA with virus injection (AAV-Dio-hM3Dq/hM4Di/mCherry) into the vmPFC of D1-Cre or D2-Cre mice. Cocaine (i.p.) was injected 30 min after CNO (i.p.) injection. Data for CNO were recorded 20–30 min after injection and cocaine were 10–20 min. D1-mChe, $n = 225$ neurons from three mice; D1-hM3Dq, $n = 296$ neurons from four mice; D1-hM4Di, $n = 333$ neurons from four mice; D2-hM3Dq, $n = 245$ neurons from three mice; D2-hM4Di, $n = 255$ neurons from three mice, two-tailed Wilcoxon test. **c, d** Schematic of two-photon Ca²⁺ imaging and statistics (minimum, maximum, and three quartiles) of normalized SSCS frequency in the FrA following microinfusion of a D1R agonist (SKF-38393, SKF) or antagonist (SCH-23390, SCH) or a D2R agonist (quinpirole, QP) or antagonist (sulpiride, sulp) into the vmPFC following i.p. cocaine in awake mice. Sal, $n = 191$ neurons from four mice; SKF, $n = 212$ neurons

from four mice; SCH, $n = 205$ neurons from four mice; QP, $n = 280$ neurons from five mice; sulp, $n = 149$ neurons from four mice. Two-tailed Wilcoxon test for paired comparisons, two-tailed Mann–Whitney test for unpaired comparisons. **e** CPP scores (**e1**, shown as preference during pre- and posttest) and locomotor distance for sensitization measurement (**e2**) under bilateral microinfusion of D1R or D2R antagonist following i.p. cocaine. Data were presented as mean ± s.e.m. CPP scores, $n = 8$ mice per group; paired $t$-test between pre- and posttest and unpaired $t$-test between posttests. Locomotor sensitization, $n = 7, 7,$ and 8 mice; in-group comparisons (between different training days) were analyzed with one-way ANOVA followed by Dunnett's multiple comparisons; multiple comparisons were performed for each group between day 2 or day 3 and day 1; unpaired two-tailed $t$-test for comparisons between SCH/sulp and sal on day 3. $P$ values were summarized in Supplementary Table 2. *$p < 0.05$; **$p < 0.01$; ***$p < 0.001$. Source data are provided as a Source Data file.

cocaine-induced hypoactivity and locomotor sensitization. This dopamine-dependent cortico-cortical pathway not only provides an example to understand how the impact of cocaine reaches distributed regions in complicated neuronal networks but also sheds new light on how cortical circuits may contribute to dopamine modulation in the physiological or pathological brain.

The third finding is the necessity of vmPFC–D1R and –D2R signaling in mediating cocaine-induced FrA hypoactivity and locomotor sensitization (Figs. 5, 6). Although both excitatory chemogenetic manipulations of vmPFC–D1 and –D2 neurons were sufficient to induce suppression of the FrA (Fig. 6b), the different percentages of suppression and different effects by antagonists (Fig. 6d) suggest different roles of vmPFC–D1R and –D2R signaling in cocaine exposure. Since connections between the vmPFC and the FrA include both excitatory and inhibitory projections, we also injected retrograde virus (Retro-CaMKIIα/vGAT-Cre) in the FrA and Cre-dependent hM3Dq/hM4Di virus in the vmPFC for chemogenetic manipulation of FrA-projecting vmPFC neurons (Glu⁺ or GABA⁺). Surprisingly, only the chemogenetic activation (DIO-hM3Dq) of FrA-projecting vmPFC GABA

⁺ neurons (Retro-vGAT-Cre) profoundly abolished the locomotor sensitization (Fig. S13), implying essential roles of GABA⁺ projections from the vmPFC to the FrA in cocaine sensitization. These findings imply the complex and manifold local circuit connections between the vmPFC and the FrA. For example, vmPFC–D1R or –D2R neurons may innervate FrA excitatory neurons via the direct projection, or indirectly through the local circuits in either the upstream vmPFC or the downstream FrA. The circuit complexity may also derive from the VTA-mPFC projections. Since VTA dopaminergic neurons also co-release glutamate or GABA with dopamine[54] and fire in various ways in response to cocaine[43,55], there is no consensus on how dopamine regulates mPFC activity[24,25]. Especially, in contrast to the studies with slice electrophysiology, in vivo studies have shown greater heterogeneity in the dopamine regulation of PFC activity[22,23,37,56], which exhibits high spatiotemporal diversity[23]. Although details of how vmPFC neurons function are obscure, our present work provides an underlying cortico-cortical pathway through which the cocaine-induced dopamine transmission indirectly suppresses FrA neural activity and thus mediates locomotor sensitization.

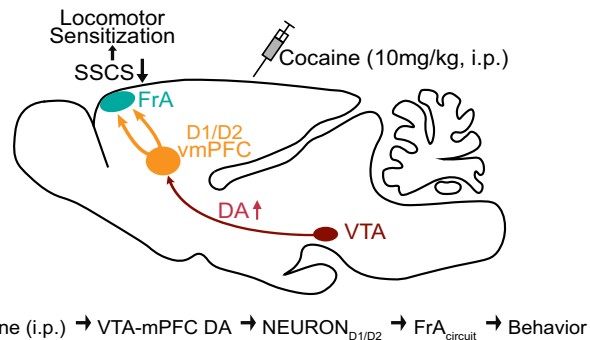

**Fig. 7 | Cartoon illustration that cocaine induces hypoactivity of the FrA and locomotor sensitization through the dopamine-dependent vmPFC–FrA pathway.** The working model is that acute cocaine exposure (10 mg/kg, i.p.) decreases FrA neuronal activity (SSCS), which is critical for cocaine-induced locomotor sensitization, through a dopaminergic VTA–vmPFC pathway relayed by a vmPFC–FrA cortico-cortical projection: Cocaine (i.p.) → VTA-mPFC DA transmission → mPFC NEURON$_{D1/D2}$ → FrA$_{circuit}$ → Behaviors.

Following the identification of this cortico-cortical circuit in the awake brain in the present study, future work may address: (1) how does vmPFC–FrA transmission function to mediate the FrA hypoactivity; (2) how does cocaine-induced dopamine abnormality regulates the activity of different types of neurons in the mPFC; (3) how does mPFC microcircuits translate the impact on different FrA neurons through mixed projections, especially in the awake state.

In summary, by using two-photon Ca$^{2+}$ imaging in the FrA of awake mice, we defined the PFC-subregion FrA as a critical region mediating cocaine-associated locomotor sensitization, and uncovered the projection of VTA$^{DA}$ neurons to the vmPFC mediates the cocaine-induced hypoactivity of FrA neurons and the locomotor sensitization: cocaine (i.p.) → VTA-mPFC DA transmission → NEURON$_{D1/D2}$ → FrA$_{circuit}$ → Behaviors (Fig. 7). This provides a brain-circuits-mechanism on cocaine-abuse of how cocaine affects the PFC and locomotor sensitization during the early onset of drug abuse, shedding new light on the mechanisms underlying dopamine-related physiological and pathological changes.

## Methods
### Animals and chemicals
Adult (3–4 months old) male C57BL/6 J mice were purchased from Charles River Laboratories; vGAT-Cre transgenic mice[57] were kindly gifted by Dr. Chen Zhang (Capital Medical University, China); DAT-Cre mice were kindly provided by Dr. Yi Rao (Peking University, China); DAT-KO mice were homozygous offspring of heterozygous DAT-Cre mice[49], which was generated by a 5′-UTR knock-in strategy to express Cre recombinase under the control of DAT promoter. DAT-CI mice were kindly gifted by Dr. Howard H. Gu (Ohio State University, USA)[3], which is a knock-in mouse line expressing functional DATs with triple mutations that are insensitive to cocaine; D1-Cre (Drd1a-Cre) and D2-Cre (Drd2-Cre)[58] and Ai9 mice[59] were gifts from Dr. Yousheng Shu (Fudan University, China). All mice were housed in the Animal Center of Peking University at $22 \pm 2\,°C$ and 50–60% relative humidity with a 12-h light/dark cycle and provided with food and water ad libitum. The use and care of animals were approved and directed by the Animal Care and Use Committee of Peking University and the Association for Assessment and Accreditation of Laboratory Animal Care.

All chemicals were from Sigma-Aldrich unless stated otherwise. Cocaine hydrochloride was from Qinghai Pharmaceutical Factory (China), GBR-12783 dihydrochloride was from Shanghai Macklin Biochemical (China), and SKF-38393 hydrobromide and SCH-23390 hydrochloride were from Tocris Bioscience (United Kingdom). Drugs

for intraperitoneal injection were dissolved in sterile saline (0.9% NaCl).

### Preparation for head-fixed two-photon Ca$^{2+}$ imaging
The fluorescent Ca$^{2+}$ indicators Cal-520[40] and GCaMP6s[41] were used[32,60] with minor modifications. All surgical instruments were sterilized before experiments and body temperature was maintained using a heating pad. Mice were anaesthetized with Avertin (250 mg/kg, i.p.) and a custom-made chamber was fixed to the skull with cyanoacrylate and dental cement. After 3 days of recovery, mice were trained to adapt to head fixation for 3 days (1–3 h per day). On the day of imaging, mice were briefly anaesthetized with 1–1.5% isoflurane in oxygen for craniotomy with a high-speed drill (RWD). Then the frontal association cortex (FrA: AP + 2.8 mm; ML + 1.0 mm) was exposed through a cranial window (1.5 × 1.5 mm) for multicell bolus loading of Cal-520 AM (500 μM; AAT Bioquest). Cal-520 AM was initially dissolved in DMSO with 20% Pluronic F-127 and then diluted with loading solution (mM): 150 NaCl, 2.5 KCl, 10 HEPES, pH 7.3–7.4[32,40]. Borosilicate pipettes (2–3 MΩ) were used to inject Cal-520 AM into the FrA by air pressure (0.02 MPa, 1 min). Before covering with a glass coverslip, the exposed brain was perfused with artificial cerebral spinal fluid (ACSF, in mM): 125 NaCl, 4.5 KCl, 26 NaHCO$_3$, 1.25 NaH$_2$PO$_4$, 2 CaCl$_2$, 1 MgCl$_2$, 10 glucose, pH 7.3–7.4 saturated with 95% O$_2$/5% CO$_2$[61]. Images were captured 1.5 h after dye loading.

GCaMP6s was used to observe excitatory and GABAergic neurons in the FrA through viral injection of AAV-CaMKIIα-GCaMP6s (AAV2/9-CaMKIIα-GCaMP6s -WPRE-pA, BrainVTA) in C57BL/6 J mice, or AAV-Flex-GCaMP6s in vGAT-Cre transgenic mice. After viral injection (method below), a chronic glass cranial window[60] was immediately created above the FrA and a custom-made chamber was glued to the skull using dental cement for subsequent head-fixed imaging (3–4 weeks later). Before imaging, mice were trained for 3 days to adapt to head fixation as described above.

### Two-photon Ca$^{2+}$ imaging
A two-photon laser scanning microscope (FV1200MPE, Olympus) was used to record Ca$^{2+}$ transients in the FrA. The excitation light (920 nm for GCaMP6s and 830 nm for Cal-520) was produced by a mode-locked Ti:Sa laser (Mai-Tai DeepSee, Spectra Physics) through a 25× water-immersion objective (1.05 NA, Olympus). Fluorescent Ca$^{2+}$ signals from individual neurons 100–200 μm beneath the cortical surface were imaged at 7.75 Hz (256 × 256 pixels, 254.5 × 254.5 μm). To assess the population activity of stochastic neuronal firing, the averaged frequency of SSCSs events collected from all visualized neurons in 6.45 min were quantified as SSCS frequency (events/min/neuron) during the basal (before drug injection) or the drug-treated state (10–20 min after i.p. injection). Two-photon images were processed and analyzed using ImageJ (v1.51t, NIH, USA) and custom-written software in MatLab (2016b, MathWorks). Each SSCS was automatically detected as an individual Ca$^{2+}$ transient where the somatic fluorescence change (dF/F$_0$ = (F − F$_0$)/F$_0$) was >3 times the standard deviation of the baseline (F$_0$)[32]. Regions of interest (ROIs) were manually identified to extract the trace of averaged fluorescence intensity (F) of each soma and F$_0$ was the averaged baseline intensity calculated with an iterative algorithm subtracting the Ca$^{2+}$ spike with high intensity. Glial cells were excluded when ROIs were determined based on the morphology and dynamics of the Ca$^{2+}$ transients.

### Viral injection
Mice were anaesthetized with Avertin (250 mg/kg, i.p.) and fixed on a digital stereotaxic instrument (RWD) for viral injection. The eyes were protected with ophthalmic ointment. Hair was removed from the head and exposed skin was scrubbed with betadine. After the incision of the skin, all coordinate measurements for different locations were made relative to the bregma unless stated otherwise. Exposed tissue was

kept wet with sterile saline (0.9% NaCl) in gel foam. The skull was thinned and cracked with the tip of a 1-mL syringe needle without damage to the cortex. The indicated viruses were injected under air pressure into the targeted area using borosilicate glass pipettes with an inner tip diameter of 9–10 µm. The injection speed was controlled at 20–30 nL/min and left for >5 min before slowly withdrawing the pipette. The skin incision was closed with nylon sutures if there was no craniotomy or chamber fixation. Animals were maintained on a heating pad until fully recovered.

The AAV viruses used in this study were: AAV-CaMKIIα-GCaMP6s (AAV2/9-CaMKIIα-GCaMP6s-WPRE-pA, $2 \times 10^{12}$ vg/mL, BrainVTA), 200 nL, unilaterally injected into the FrA (AP + 2.80 mm; ML + 1.00 mm; 0.2–0.3 mm from the dura surface); AAV-Flex-GCaMP6s (AAV2/9-Syn-Flex-GCaMP6s-WPRE-SV40, $2 \times 10^{12}$ vg/mL, Penn Vector Core), 200 nL, unilaterally injected into the FrA of vGAT-Cre mice; AAV-CaMKIIα-hM3Dq (AAV2/9-CaMKIIα-hM3Dq-mCherry- WPRE-pA, $6.0 \times 10^{12}$ vg/mL, BrainVTA), AAV-CaMKIIα-hM4Di (AAV2/9-CaMKIIα-hM4Di-mCherry-WPRE-pA, $5.8 \times 10^{12}$ vg/mL, BrainVTA) and AAV-CaMKIIα-mCherry (AAV2/9-CaMKIIα-mCherry-WPRE-pA, $4.1 \times 10^{12}$ vg/mL, BrainVTA), 200 nL/side, bilaterally injected into the FrA; AAV-Retro-hSyn-EYFP (AAV2/R-hSyn-EYFP-WPRE-pA, $5.0 \times 10^{12}$ vg/mL, BrainVTA), 200 nL, unilaterally injected into the FrA; AAV1-hSyn-Cre ($1.4 \times 10^{13}$ vg/mL, OBiO Technology), 100 nL/site, unilaterally injected into the vmPFC (AP + 1.70 mm; ML + 0.30 mm; DV −3.30/−3.00 mm) of Ai9 mice; AAV-Retro-CaMKIIα-EYFP (AAV2/R-CaMKIIα-EYFP-WPRE-pA, $5.0 \times 10^{12}$ vg/mL, BrainVTA), 200 nL, unilaterally injected into the FrA of D1-/D2-Cre × Ai9-crossed mice; AAV-Retro-vGAT-EYFP (AAV2/R-vGAT1-EYFP-WPRE-pA, $5.1 \times 10^{12}$ vg/mL, BrainVTA), 200 nL, unilaterally injected into the FrA of D1-/D2-Cre × Ai9-crossed mice; AAV-Flex-ChR2-EYFP (AAV2/9-EF1α-Flex-hChR2(H134R)-EYFP-WPRE-gH, $4.9 \times 10^{12}$ vg/mL, Penn Vector Core), 200 nL/side, bilaterally injected into the vmPFC (AP + 1.75 mm; ML ± 0.30 mm; DV −3.05 mm) of D1-/D2-Cre mice. AAV-Dio-hM3Dq (AAV2/9-EF1α-Dio-hM3Dq-mCherry-WPRE-pA, $5.0 \times 10^{12}$ vg/mL, BrainVTA), AAV-Dio-hM4Di (AAV2/9-EF1α-Dio-hM4Di-mCherry-WPRE-pA, $5.2 \times 10^{12}$ vg/mL, BrainVTA) and AAV-Dio-mCherry (AAV2/9-EF1α-Dio-mCherry -WPRE-pA, $5.1 \times 10^{12}$ vg/mL, BrainVTA), 200 nL/side, bilaterally injected into the vmPFC (AP + 1.75 mm; ML ± 0.30 mm; DV −3.05 mm) of D1-/D2-Cre mice. Viruses or drugs were aliquoted and stored at −80 °C before use. For all experiments involving stereotaxic injections, animals targeting the wrong location were excluded after verification.

## Behavioral tests

Mice were trained and tested in a custom-built acrylic arena inside an individual room with control of light, temperature, odor, and noise. The locations and locomotion of mice were automatically captured by a video tracking system LabState5.10 (Anilab Software & Instruments)[32] to measure cocaine-induced CPP and locomotor sensitization[43]. The arena was designed with three interconnected chambers: two lateral chambers A/B (25 cm × 25 cm for each) and a middle chamber (8 cm × 25 cm). The floor of chamber A was equipped with parallel rods, chamber B with a square grid, and the middle chamber with a smooth floor. As the timeline in Fig. 1l shows, on day 0 (pretest), mice were transferred to the middle chamber and allowed free access to the three chambers for 15 min. There was no significant basal preference for chamber A or B before conditioning. During days 1–3 (conditioning), mice were repeatedly conditioned with saline and cocaine injections (i.p.) paired to opposite lateral chambers while their locomotion was recorded. Each morning, a mouse was confined to a given lateral chamber (A or B, stochastically) for 30 min immediately after cocaine (or saline in alternation), and confined to the other side immediately after saline injection (or cocaine accordingly) in the afternoon (6 h later). Mice with paired saline administration served as controls. To test the curative effect of FrA excitation, the AAV-CaMKIIα-hM3Dq mice were used and CNO (0.5 mg/kg, i.p.) was injected 30 min before

cocaine (10 mg/kg, i.p.) injection (or a random saline injection in the control group). The percentages of mice receiving cocaine injection in the morning or afternoon and those paired to chamber A or B were equal to achieve a counterbalance design. On day 4 (posttest), mice were tested as on day 0. The CPP score was defined as the time (T) differences of preference in the cocaine-paired chamber between the posttest and pretest: CPP score = $(T_{coc} − T_{sal})_{posttest} − (T_{coc} − T_{sal})_{pretest}$. Cocaine-induced locomotor sensitization was assessed as the distances traveled in 30 min after cocaine injection during the 3 conditioning days. Behavioral tests were performed 4–6 weeks after the viral injection.

## Lesion of dopamine neurons

The neurotoxins 6-OHDA[62,63] and MPTP[62,64] were used to deplete dopamine neurons/projections with slight modifications. MPTP was repetitively injected for 5 days (30 mg/kg, i.p. daily) to preferentially induce the loss of SNc dopaminergic neurons and the projections mainly targeting the CPu. The stereotaxic microinjection of 6-OHDA (2.5 µg/µL, in saline with 0.2 mg/ml ascorbic acid) was applied for the local depletion of dopaminergic terminals or somata in the targeted regions. The 6-OHDA was bilaterally injected (200 nL per site) into the FrA (in mm: AP + 2.80; ML ± 1.00; 0.2–0.3 from the dura surface), the VTA (site1/2: AP −3.20; ML ± 0.40; DV −4.40; site2/2: AP −3.60; ML ± 0.40; DV −4.15), the SNc (site1/2: AP −3.30; ML ± 1.45; DV −4.00; site2/2: AP −3.30; ML ± 1.45; DV −4.60), the vmPFC (AP + 1.70; ML ± 0.30; DV −3.20/−2.90), and the BLA (AP −1.15; ML ± 2.90; DV −4.55). The surgery for two-photon imaging preparation was the same as above and performed immediately after microinjection. One week after 6-OHDA injection (but 21–28 days in the FrA), mice were used for two-photon Ca$^{2+}$ imaging, and were then sacrificed to verify the loss of dopaminergic neurons/terminals with TH-staining (details below).

## Immunofluorescence

Mice were anaesthetized with Avertin (250 mg/kg, i.p.) and transcardially perfused with 4% paraformaldehyde (PFA) in PBS. Brains were postfixed in 4% PFA over 12 h and dehydrated in 10, 20, and 30% sucrose over 72 h for cryoprotection. Then frozen sections (40 µm; CM1950, Leica) were collected for permeabilization (0.3% Triton X-100 in PBS) and blocking (2% bovine serum albumin in PBS). After that, the sections were incubated with rabbit anti-TH primary antibody (1:500; AB152, Millipore) or mouse anti-GAD67 primary antibody (1:500; MAB5406, Millipore) overnight at 4 °C. After three washes, the sections were incubated with secondary antibodies (1:500; donkey anti-rabbit 594, A21207 or donkey anti-mouse 488, A21202; Invitrogen) for 1 h and DAPI for nuclear staining at room temperature. The sections were mounted on slides with 50% glycerol and kept at −20 °C. Immunofluorescence images were captured using a laser scanning confocal microscope (LSM710, Carl Zeiss) and paired sections were imaged and processed at the same settings. ImageJ (v1.51t, NIH, USA) and Adobe Illustrator (v2019-23.1.0, Adobe Systems) were used for image processing.

## Amperometric dopamine recording

Carbon fiber electrodes (CFEs, 7 µm in diameter) were used for in vivo electrochemical amperometric recording of evoked dopamine overflow[65]. In brief, mice were placed into a stereotaxic instrument (Narishige) under anesthesia (Avertin, 250 mg/kg, i.p.) and craniotomies were made for CFE recording in the vmPFC (in mm: AP + 1.75; ML + 0.30; DV −3.30), bipolar-electrode stimulation in the ipsilateral medial forebrain bundle (AP + 1.00; ML + 1.40; DV −5.20 to −5.60), and the insertion of an Ag/AgCl reference electrode into the contralateral visual cortex (AP −3.50; ML −2.30; DV −1.00). According to our lab's works[65,66] and others' works[67,68], it's well-established that it's easier to get stable evoked terminal signals from the projection bundles in the

MFB, which were confined to a limited region, but not from the somata region in the VTA which are usually more dispersed. The holding potential of the CFE was maintained at 780 mV under the control of a patch-clamp amplifier (low-pass filtered at 50 Hz; PC2C, INBIO) and MBA-1 software (INBIO). The electrical stimulus was generated by an isolator (A395, WPI) as a train of biphasic square-wave pulses (0.6 mA, 1 ms duration). CFE signals were recorded when stable during the basal state and 10–20 min after cocaine injection (10 mg/kg, i.p.). The amperometric traces were processed and analyzed using IGOR Pro 6.37 (WaveMetrics).

## Micro-infusion

Cannula-guided microinfusion of drugs was combined with two-photon $Ca^{2+}$ imaging or behavioral tests in awake mice to investigate dopamine signaling in the vmPFC and BLA. For two-photon calcium imaging, a single guide cannula (O.D. 0.41 mm) was inserted into the vmPFC (in mm: AP + 1.30; ML + 0.30; DV −2.70) at an angle of 53° to the horizontal plane to leave space for the objective (40×/0.80 W/3.5 WD, Olympus). For behavioral tests, double guide cannulae (O.D. 0.41 mm) were inserted into the vmPFC (in mm: AP + 1.75; ML ± 0.30; DV −2.50) vertically. The cannula implantation in the BLA was similar, except that it was vertically inserted at AP −1.15; ML + 2.90; DV −4.10. The guide cannula was secured with dental cement and a dummy cannula was inserted to prevent clogging. The surgery for two-photon imaging was the same as above and performed immediately after cannula surgery. After recovery for 7 days, an injector cannula (O.D. 0.21 mm) with a 0.5-mm extension beyond the tip of the guide cannula was used for microinjection. The injector cannula was connected to a 10-μL Hamilton syringe controlled by a microinfusion pump (KD Scientific). Cocaine hydrochloride (40 mg/mL), dopamine hydrochloride (40 mg/mL), SKF-38393 hydrobromide (5 mg/mL), SCH-23390 (2 mg/mL), quinpirole hydrochloride (5 mg/mL), and sulpiride hydrochloride (2 mg/mL) were dissolved in 0.9% saline and infused at 0.2 μL/min for 1 min. If combined with imaging, the microinfusion was applied 5 min after i.p. injection of cocaine and images were recorded 5 min after microinfusion. If combined with behavioral tests, the microinfusion was applied 1–3 min after i.p. injection of cocaine and the animals were transferred to the behavioral chambers 1 min after the end of micro-infusion. The infusion sites were verified by infusion of Chicago sky blue and sectioning after imaging. Mice with incorrect microinfusion locations were excluded.

## Neural tracing

We used retrograde and trans-synaptic anterograde AAV to identify afferents of the FrA and the types of FrA neurons receiving the terminals of vmPFC efferent neurons. Retrograde AAVs (AAV-Retro-hSyn-EYFP, AAV-Retro-CaMKIIα-EYFP, and AAV-Retro-vGAT-EYFP) were unilaterally injected into the FrA. Sequential coronal sections of mouse brains were acquired as above and fluorescent images were analyzed referring to the atlas[69]. The anterograde trans-monosynaptic tracing virus (AAV1-hSyn-Cre)[70] was unilaterally injected into the vmPFC of Ai9 (tdTomato) mice. Coronal sections of the FrA were immunostained with anti-GAD67 antibody for the identification of neuronal types receiving vmPFC inputs. The FrA was roughly divided into different layers (layer I, 0–100 μm; layers II/III, 100–500 μm; layers V/VI, 500–1000 μm) based on related studies[34] and the very low density of somata in layer I. Soma numbers were counted in an area of 500 × 500 μm in vmPFC slices and 1 mm ×1 mm in FrA slices.

## Electrophysiology in brain slices

Coronal brain slices were prepared for patch-clamp recording[71]. Briefly, mice were anesthetized with urethane (1.5 g/kg, i.p.) and transcardially perfused with 10 mL ice-cold cutting solution containing (mM): 110 choline chloride, 2.5 KCl, 1.3 NaH$_2$PO$_4$, 0.5 CaCl$_2$, 7 MgCl$_2$, 25

NaHCO$_3$, 10 glucose, 1.3 L-ascorbate, 0.6 Na-pyruvate (pH 7.3–7.4, saturated with 95% O$_2$/5% CO$_2$). The brains were cut at 250 μm on a vibratome (VT1200S, Leica) in an ice-cold cutting solution and incubated in ACSF for 30 min at 35 °C and another 30 min at room temperature (22–25 °C) before recording. The ACSF contained (mM): 125 NaCl, 2.5 KCl, 1.3 NaH$_2$PO4, 2 CaCl$_2$, 1.3 MgCl$_2$, 25 NaHCO$_3$, 10 glucose, 1.3 L-ascorbate, 0.6 Na-pyruvate (pH 7.3–7.4, saturated with 95% O$_2$/5% CO$_2$).

Slices in the recording chamber were superfused (2 mL/min) with ACSF during electrophysiological recording. A multi-channel perfusion device (MPS-1, INBIO) was used for local drug delivery (in ACSF) to the recorded cells. An upright microscope (BX51WI, Olympus) equipped with fluorescent, infrared, and differential interference contrast devices were used to visualize cells and guide patch pipettes (3–4 MΩ, borosilicate glass, WPI). We used an EPC9/2 amplifier and Pulse software (HEKA Elektronik) to obtain whole-cell patch-clamp recordings and signals were digitized at 20 kHz and low-pass filtered at 2.9 kHz. Electrophysiological data were processed with IGOR Pro 6.37 (WaveMetrics).

Whole-cell voltage-clamp (−70 mV) recording of light-evoked EPSCs and IPSCs was applied to identify the synapses between vmPFC−D1R-/D2R and FrA layer II/III neurons. AAV-Flex-ChR2-EYFP was injected into the vmPFC of D1-/D2-Cre mice (Fig. S12) 4–6 weeks before recording. Cells in FrA slices were randomly chosen according to their pyramidal shape and layer II/III location (100–500 μm). The pipette internal solution contained (mM): 153 CsCl, 1 MgCl$_2$, 10 HEPES, 4 Mg-ATP, and 3.3 QX314 (pH 7.3 adjusted with CsOH). Photostimulation (1 ms, 0.1 Hz) was delivered by a 473-nm laser device (VD-IIIA, Beijing Viasho Technology) under the control of an EPC9/2 amplifier. IPSCs were recorded in the presence of 50 μM D-AP5 (D(-)−2-amino-5-phosphonovaleric acid, an antagonist of NMDA receptors) and 10 μM CNQX (6-cyano-7-nitroquinoxaline-2,3-dione, an antagonist of AMPA receptors). EPSCs were recorded in the presence of 100 μM PTX (picrotoxin, an antagonist of GABA receptors).

Whole-cell current-clamp (0 pA) recording of action potentials was used to verify the excitatory modulation of FrA layer II/III pyramidal neurons by chemogenetic activation. AAV-CaMKIIα-hM3Dq-mCherry was injected into the FrA. hM3Dq$^+$ cells were identified through the fluorescence of mCherry. The pipette internal solution contained (in mM): 130 K-gluconate, 10 KCl, 2 MgCl$_2$, 2.5 Mg-ATP, 0.25 Na-GTP, 10 HEPES, and 0.4 EGTA (adjusted to pH 7.3 with KOH). Action potentials were stimulated by current injection (200 pA, 500 ms) and recorded before and after perfusion with CNO (10 μM).

## Statistical analysis

Statistical analyses were made using Prism 7 (GraphPad Software). The sample size was not predetermined but similar to those in similar studies[32,43,56,72]. All $t$-tests and non-parametric tests were two-tailed and data were presented as the mean ± s.e.m for $n < 12$ with data points plotted or box-and-whisker plots for $n > 12$ (minimum to maximum with three quartiles marked). Normality was tested for all two-photon $Ca^{2+}$ imaging data by the D'Agostino-Pearson omnibus test. Non-parametric data were analyzed with the Wilcoxon test for paired comparisons or the Mann–Whitney test for unpaired comparisons and the others were analyzed with paired or unpaired Student's $t$-test. For grouped analyses, one-way ANOVA followed by Dunnett's multiple comparisons or two-way ANOVA followed by Bonferroni's or Tukey's multiple comparisons were used to make comparisons. All the detailed statistical methods and results are listed in the corresponding legends. All animals and samples that were successfully tested were included in our analysis, and at least three biologically individual animals were used for repetition in each experiment. Significant differences were accepted at $p < 0.05$ and thresholds were placed at *$p < 0.05$, **$p < 0.01$, and ***$p < 0.001$.

## Reporting summary

Further information on research design is available in the Nature Portfolio Reporting Summary linked to this article.

## Data availability

All relevant data that support this study are available from the corresponding authors upon request. Source data are provided with this paper.

## Code availability

The code used in this study is freely available for academic or non-commercial users via https://github.com/lunwangm/2PMI.

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

## Acknowledgements

We thank Drs. Chen Zhang (Capital Medical University, China) for vGAT-Cre mice, Yi Rao (Peking University, China) for DAT-Cre mice, Howard H. Gu (Ohio State University, USA) for DAT-CI mice, Yousheng Shu (Fudan University, China) for D1-/D2-Cre and Ai9 mice, Mr. Xi Wu (Peking university) for cartoon in Fig. 1a, and Dr. Iain C. Bruce (Peking University) for reading the manuscript. This work was supported by the National Natural Science Foundation of China (31930061, 21790394, 82000757, 31761133016, 31821091, 31330024, 31171026, 31327901, 32171233, and 21790390) to Z.Z., the National Key Research and Development Program of China (2016YFA0500401) to Z.Z., the National Natural Science Foundation of China (32171233 and 31670843) to C.W., the Natural Science Foundation of Shaanxi Province of China (2023-ZDLSF-23, 2021TD-37, and 2019JC-07) to C.W., the National Natural Science Foundation of China (82101568) to F.Z., STI2030-Major Projects(2021ZD0203900) to F.Z., and the China Postdoctoral Science Foundation (2021M690219) to L.W. Dr. Lun Wang was supported in part by a Postdoctoral Fellowship from the Peking-Tsinghua Center for Life Sciences.

## Author contributions

Z.Z. and L.W. designed the research. L.W., M.G., and Q.W. performed experiments and analyzed data. L.Su., M.Y., S.M., C.L., L.Sh., Y.L., B.Z., S.S., G.C., and J.L. performed experiments. Z.Z., L.W., M.G., Q.Z., F.Z., and C.W. wrote the paper.

## Competing interests

The authors declare no competing interests.
