## [Peer Review File · Nature Communications]

Cocaine induces locomotor sensitization through a dopamine-dependent VTA-mPFC-FrA cortico-cortical pathway in male miceREVIEWER COMMENTS

Reviewer #1 (Remarks to the Author):

In this current study, Wang and colleagues investigated the role of frontal association cortex (FrA) neurons in cocaine-induced locomotor sensitization. Utilizing transgenic mice, two-photon Ca²⁺ imaging, chemogenetics, electrophysiology and biochemistry approaches, the authors report that acute cocaine induced a hypoactivity of FrA neurons, which is necessary for cocaine to induce locomotor sensitization. The cocaine-induced hypoactivity of FrA neurons was DAT dependent, and they further found that FrA neurons were regulated by the projection of VTADA neurons to the vmPFCD1R neurons, demonstrating a cortico-cortical circuit bridging dopamine transmission.

Overall, this is a very well-written manuscript, the results are extremely interesting, potentially provide a novel sight into the role of VTA-vmPFC-FrA circuit. Most of the experiments are well designed and executed. However, as compared to their solid data from Ca²⁺ imaging, pharmacological and electrophysiological studies, some circuit-related behavioral measurements are missed, which is useful to support the conclusion of this interesting study. The comments outlined below should be thoroughly addressed by the authors.

Major comments:

1) The hM3Dq data are impressive, which demonstrate that the hypoactivity of FrA neurons is necessary for cocaine-induced locomotor sensitization, but the validation data of CNO effect on SSCS frequency (Fig S3c) showed a higher frequency as compared to control. It would be useful to investigate the inhibitory DREADDs to mimic the acute cocaine-induced hypoactivity of FrA neurons and see if the inhibitory DREADDs-induced FrA hypoactivity promotes locomotor sensitization with a subthreshold dose of cocaine.

2) It is unknown how many hM3Dq-manipulated FrA neurons have direct connection with vmPFCD1R neurons. It would be important to manipulate the vmPFC-FrA circuit and invest the effect of such manipulation on cocaine-induced locomotor sensitization. These experiments would be important to support their conclusion.

3) Additionally, using an elegant approach, the authors demonstrate that vmPFCD1R is an important component that innervates with VTADA inputs and responsible for the cocaine-induced FrA neuronal hypoactivity. However, whether the VTADA-vmPFCD1R circuit is the key element that contributes to cocaine-induced locomotor sensitization behavior may need to be proved.

4) To prove that it was the VTA dopaminergic neurons, but not the SNc neurons that play essential roles in mediating cocaine-induced hypoactivity in the FrA neurons, the authors examined FrA activity after a depletion of VTA and SNc-CPu dopaminergic neurons by using 6-OHDA and MPTP approaches, respectively (Line 203-208). Since the MPTP treatment was a systemic administration, can the MPTP metabolite also be transported by DAT into the dopaminergic cells in VTA? Whether the FrA activity change

was a concurrent result of VTA and SNc dopaminergic neuronal depletion? It would be better to apply a same treatment (6-OHDA lesion) to deplete SNc region and record FrA activity, as well as provide both the SNc and CPu TH-staining results (Fig S6).

5) By using the D1-Cre;Ai9 mice and viral tools, the authors showed that vmPFC D1R neurons were involved in mediating cocaine-induced hypoactivity of the FrA neurons. It seems that D1 and D2 receptors are both expressed in pyramidal neurons and GABAergic interneurons in mPFC. Although D1R are usually considered exerting an excitatory effect while D2R producing an inhibitory effect, recent study has demonstrated that D2R signaling in the mPFC is often excitatory and these receptors signaling usually complex in mPFC. It would be interesting to know whether the D2R also play a crucial role in mediating the neuronal hypoactivity in FrA?

Minor comments:

- 1) Color code in Fig 1 is unclear. Information for each color code in Fig 1 should be provided.
- 2) The TH-positive signal in FrA of sham group that provide in Fig.S7 was weak, it was difficult to tell the difference between sham and 6-OHDA group. Those figures should be replaced with better quality ones.
- 2) In Fig 4d, it is confusing why the authors illustrated the placement of stimulation electrode was in MFB, not in the VTA. In addition, some of the schematic illustrations appeared in figure legends were "VTA" and some were presented as "DAN". Clearer information should be provided about the full name of "DAN" when it appears first time in the manuscript.
- 3) The pulse number used in Fig.4e "80 p, 80 Hz" seems a very strong stimulation. More explanation would be helpful about how it was selected.
- 4) Detail information should be provided about DAT-KO and DAT-CI, SSCS frequency, statistic analyses.

Reviewer #2 (Remarks to the Author):

Wang and coworkers provide intriguing new data implicating the frontal associative cortex in cocaine-induced locomotor sensitization. The authors use a variety of powerful techniques to show decreased FrA neuronal activity after cocaine treatment, implicate these neurons in sensitization, and assess the role of dopaminergic projections to vmPFC in the FrA actions. In general the experimental design is strong and the conclusions are well supported by the data. The use of both a calcium dye and GCaMP in the imaging experiments is a nice feature of the study, as the dye allows for better temporal resolution while GCaMP allows for cell-type specific analysis. The lack of effect on CPP is interesting, as it suggests a proscribed role for the FrA. There are aspects of experimental design and analysis that could be

improved and/or require additional information. Overall, the findings are new, convincing, and have relevance for understanding the undesirable effects of cocaine.

Major Comments

1. The analysis of the sensitization data in the DREADD experiment does not appear to include a comparison across groups, only a trial-dependent difference in the control group and no such change in the DREADD group. To conclude that the two groups differ it is necessary to perform a between-groups comparison, preferably with a mixed-model ANOVA of some sort. The graph showing these data, figure 1n, also needs labels to designate the different groups.
2. The authors should exercise a bit more caution in discussing locomotor sensitization as an indicator of substance use disorders. While locomotor sensitization may be related to incentive sensitization or incubation of craving, the link between these behaviors is not always clear. Discussing future studies using other behavioral measures would be useful in this context.
3. The authors need to quantify the loss of dopaminergic fibers in the FrA and vmPFC following local 6-OHDA lesions. The main concern is that the lesions in FrA may not be of sufficient magnitude to prevent signaling important for sensitization.
4. The experiments showing the efficacy of hM3Dq in altering FrA activity needs a group with no DREADD expression to ensure that CNO has no effects on physiology in the absence of the DREADD.
5. The rationale for the focus on D1-expressing vmPFC neurons is not explained, although it appears to arise from the involvement of dopamine in the FrA responses. However, it is unclear why vmPFC neurons expressing other dopamine receptors (e.g. D2) were not examined. It is recognized that the authors could not examine every neuronal subpopulation, but some explanation of how the focus on D1-expressing neurons was chosen would be helpful.
6. For the CPP experiment, it is good that the authors recorded locomotor activity during the test, but they should also include total time in each chamber before and after conditioning in addition to the normalized CPP score. These data could be presented in the supplement, especially if there is no effect of the DREADD treatment on this measure.
7. Was 2p excitation used for both dye- and GCaMP-based calcium imaging? If so, is the 920 nm wavelength optimal for GCaMP excitation?
8. Figure 4e, the stimulus induced transients appear to be longer lasting in the presence of cocaine, and the authors mention decreased uptake when discussing these results. However, the authors do not present the quantification of this measure. Also, the traces shown in this figure do not decay back to baseline levels so it is not clear if measures of transient duration or area under the curve would be accurate. This group has previously provided evidence that dopamine mediates these amperometric

responses in rat (e.g. with voltammetry), but is there comparable evidence in mouse?

Minor Comments:

a. The title of the paper is inaccurate. Cocaine produces locomotor sensitization and the effects on FrA, but these effects are actually mediated by the neural mechanisms described in the paper.

b. The term addiction is now thought to be pejorative, and most publications are now switching to substance use disorders. The authors might consider making this change as well. In addition, on line 52 the authors imply that rodents can be addicts, but it is difficult to judge if this is the case given that many of the criteria for SUDs require behaviors that can be assessed in humans but not so readily in rodents.

c. Does the FrA have a role in normal locomotion? The data in figure 1n suggests that it probably does not, but it might be worth discussing this topic.

d. It would be interesting to know how the calcium signals recorded with GCaMP6s compare to those recorded with the calcium-sensitive dye. Were they slower and/or less numerous with GCaMP? The underlying question is if GCaMP6s can report spiking activity comparable to that observed with the dye.

e. What was the temperature during the brain slice recordings? Could the local drug perfusion alter temperature, i.e. by introducing a room temp solution in a heated bath?

Reviewer #3 (Remarks to the Author):

Wang et al identify a novel and interesting circuit underlying the capacity of cocaine to promote locomotor sensitization in mice. Using in vivo 2P calcium imaging, they show that glutamate neuron activity in frontal association cortex (FrA) is strongly reduced for tens of minutes after an injection of cocaine. When these cocaine-induced reductions in FrA activity are counteracted by optogenetic stimulation, locomotor sensitization is not evident, as it is in control mice, following two subsequent injections of cocaine. With clever follow-up experiments, the authors show that the cocaine-induced change in FrA activity is dependent on dopamine and D1 receptor signaling in the ventromedial prefrontal cortex (vmPFC), which they show sends direct, albeit mixed, projections to FrA.

Overall, the manuscript is interesting and novel. With modern techniques, the authors provide multiple converging lines of evidence to support their conclusion that cocaine's influence on FrA activity is indirect, mediated by dopamine and D1 receptor signaling in the vmPFC. Although the work is thorough and convincing, there are two considerable limitations that should be addressed prior to publication.

Major:

1. The introduction should be rewritten to improve general clarity and include more sophisticated use of psychological language. Specifically,

- a) mice should not be described as addicted, and the use of “addicts” should be omitted completely;
- b) undefined language like “cocaine-adaptive behaviours” should be avoided or specified;
- c) locomotor sensitization should not be equated with incentive saliency without a clear description of the term and citations to support the claim.

2. The purported mechanism through which increased D1 receptor signaling in vmPFC produces a sustained reduction in FrA activity is unclear. The authors acknowledge this point without really addressing it. What microcircuits might be involved? Can the authors suggest a mechanism, either by more clearly describing the existing literature or by conducting additional experiments? The sizeable literature on how dopamine affects vmPFC activity should be discussed in more detail, as it may provide insight into how dopamine-induced changes in vmPFC activity can diminish FrA activity. It may be helpful to describe how the effects of dopamine on vmPFC activity seem to be dependent on ongoing network activity (as reviewed in Seamans and Yang, 2004, *Progress in Neurobiology*). Alternatively, the authors could provide evidence that vmPFC-to-FrA projections can actually reduce FrA activity. They could test whether chemogenetic activation of neurons in the vmPFC (whether all of them or just the glutamate or GABA subpopulations) is sufficient to reduce FrA activity. These experiments would shed light on the underlying mechanism by which dopamine signaling in the vmPFC reduces FrA activity.

Minor:

1. The rationale for recording FrA activity in response to cocaine is weak. It may be possible to bolster the rationale by discussing previous literature that characterizes the effects of dopamine on mPFC activity within the context of substance use disorders.

2. This statement in the introduction, “...it is not clear how and to what extent the PFC of an awake mouse is affected by acute cocaine” is dismissive of prior work in this area. It would be better to describe what actually is and is not known about the acute actions of cocaine in the PFC.

Point-by-point response to the reviewers' comments

Reviewer #1 (Remarks to the Author):

In this current study, Wang and colleagues investigated the role of frontal association cortex (FrA) neurons in cocaine-induced locomotor sensitization. Utilizing transgenic mice, two-photon Ca²⁺ imaging, chemogenetics, electrophysiology and biochemistry approaches, the authors report that acute cocaine induced a hypoactivity of FrA neurons, which is necessary for cocaine to induce locomotor sensitization. The cocaine-induced hypoactivity of FrA neurons was DAT dependent, and they further found that FrA neurons were regulated by the projection of VTADA neurons to the vmPFC/D1R neurons, demonstrating a cortico-cortical circuit bridging dopamine transmission. Overall, this is a very well-written manuscript, the results are extremely interesting, potentially provide a novel sight into the role of VTA-vmPFC-FrA circuit. Most of the experiments are well designed and executed. However, as compared to their solid data from Ca²⁺ imaging, pharmacological and electrophysiological studies, some circuit-related behavioral measurements are missed, which is useful to support the conclusion of this interesting study. The comments outlined below should be thoroughly addressed by the authors.

Response: Thanks for your encouragement on this work and your criticism on behavioral assessment. Following your advice, we performed new experiments (**Figs. R1-R6** for responses) and revised the manuscript to address your comments on the behavioral and other issues, including the effect of inhibitory regulation of FrA on subthreshold cocaine-induced sensitization (**Fig. R1**), the effect of vmPFC-FrA manipulation on behaviors (**Fig. R3**), the effect of VTA-vmPFC manipulation on behaviors (**Fig. R4**) and FrA activity (**Fig. R6**), and the roles of vmPFC-D2R (**Figs. R2, R6**). For **Figs. R2** and **R6**, only the data about D2 receptor and **Fig. R6b** are new. All these new data are included in the revised manuscript.

Major comments:

1) The hM3Dq data are impressive, which demonstrate that the hypoactivity of FrA neurons is necessary for cocaine-induced locomotor sensitization, but the validation data of CNO effect on SSCS frequency (Fig S3c) showed a higher frequency as compared to control. It would be useful to investigate the inhibitory DREADDs to mimic the acute cocaine-induced hypoactivity of FrA neurons and see if the inhibitory DREADDs-induced FrA hypoactivity promotes locomotor sensitization with a subthreshold dose of cocaine.

Response: Following your advice, to test whether the inhibitory DREADDs (hM4Di) promote subthreshold cocaine-induced locomotor sensitization, we have added new experiments by first examining the cocaine dose-dependent behaviors and selected 2.5 mg/kg as the subthreshold dose [**Fig. R1a** (new **Fig. S6a**)], which is only able to induce significant locomotor distance sensitization on day 3 but not on day 2. Then, we applied the same protocol with Fig. 1k-n but using hM4Di instead of hM3Dq to investigate the role FrA inhibition in cocaine effect. The data

show that locomotor sensitization of hM4Di-coc mice were indeed facilitated compared with the control (mCherry-coc) group [Fig. R1b (new Fig. S6b)]. The locomotor distance of hM4Di-coc mice was significantly increased on the day 1 to a similar level as day 3 and thus didn't express further sensitization across the 3 days examination [Fig. R1b (new Fig. S6b)]. Thus, although the inhibitory hM4Di worked, compared to excitatory hM3Dq, it produced an enhanced hyperactivity and thus no further facilitation was observed.

Fig. R1 (new Fig. S6) Inhibitory modulation of the FrA by hM4Di changes the subthreshold cocaine-induced locomotor sensitization. (a) Statistics of CPP scores (a1) and locomotor sensitization (a2) by different doses of cocaine (10, 5 or 2.5 mg/kg) for intact C57BL/6J mice, $n = 8$ per group. Unpaired *t*-test for CPP scores comparisons between cocaine and saline groups. In-group comparisons (between different training days) of locomotor distance were analyzed with one-way ANOVA followed by Dunnett's multiple comparisons: sal, $F(1.996, 13.97) = 0.4961$, $p = 0.62$; 10 mg/kg coc, $F(1.176, 8.23) = 14.25$, $p = 0.004$; 5 mg/kg coc, $F(1.748, 12.24) = 14.64$, $p < 0.001$; 2.5 mg/kg coc, $F(1.773, 10.64) = 5.062$, $p = 0.03$; multiple comparisons were performed for each group between day 2 or day 3 and day 1 with *p* values marked in the plot. Data are presented as the mean \pm s.e.m.. (b) Statistics of CPP scores (b1) and locomotor sensitization (b2) by subthreshold dose of cocaine (2.5 mg/kg) for C57BL/6J mice with AAV-hM4Di bilaterally expressed in FrA, $n = 11$ (mCher-sal), 12, 12, 12. Detailed data points of day 1 and day 3 for cocaine groups are presented in (b3). CPP scores, ordinary two-way ANOVA followed by

Bonferroni's multiple comparisons test: drug effect, $F(1, 43) = 5.192$, $p = 0.028$; virus effect, $F(1, 43) = 0.302$, $p = 0.59$; interaction, $F(1, 43) = 0.05785$, $p = 0.81$; all p values for multiple comparisons are not significant. Locomotor sensitization measurement, in-group comparisons (between different training days) were analyzed with one-way ANOVA followed by Dunnett's multiple comparisons: AAV-mCherry (sal), $F(1.484, 14.84) = 0.9339$, $p = 0.3878$; AAV-mCherry (coc), $F(1.336, 14.7) = 8.013$, $p = 0.0086$; AAV-hM4Di (sal), $F(1.91, 21.01) = 0.9565$, $p = 0.3966$; AAV-hM4Di (coc), $F(1.647, 18.12) = 0.1479$, $p = 0.8238$; multiple comparisons were performed for each group between day 2 or day 3 and day 1 with p values marked in the plot. Unpaired t -test for comparisons between cocaine groups on the day 1. Data are presented as the mean \pm s.e.m.. n.s., not significant; * $p < 0.05$; ** $p < 0.01$.

2) It is unknown how many hM3Dq-manipulated FrA neurons have direct connection with vmPFC-D1R neurons. It would be important to manipulate the vmPFC-FrA circuit and invest the effect of such manipulation on cocaine-induced locomotor sensitization. These experiments would be important to support their conclusion.

Response: Following your advice, we (1) quantified FrA layer II/III pyramidal neurons receiving vmPFC-D1R or -D2R neurons' projections. About 55.0% and 33.3% of tested neurons were positive with synaptic input currents from vmPFC-D1R and -D2R neurons' projections, respectively [Fig. R2 (new Fig. 5h-k)]; (2) combined injection of (i) retrograde virus expressing Cre (driven by $CaMKII\alpha$ or vGAT) in the FrA and (ii) Cre-dependent hM3Dq/hM4Di virus in the vmPFC for the chemogenetic manipulation of FrA-projecting vmPFC neurons (Glu^+ or $GABA^+$). Surprisingly, only the chemogenetic activation (DIO-hM3Dq) of FrA-projecting vmPFC $GABA^+$ neurons (Retro-vGAT-Cre) profoundly abolished locomotor sensitization [Fig. R3 (new Fig. S13)], implying essential roles of $GABA^+$ projections from the vmPFC to the FrA in cocaine sensitization. At the beginning, we tried to manipulate the FrA-projecting vmPFC-D1 or -D2 neurons that are glutamatergic or GABAergic, but unfortunately the virus combinations (Retro- $CaMKII\alpha$ /vGAT-Frt-DREADD and Dio-Flp in D1/D2-Cre mice) failed to efficiently label the specific neurons after several trials in our lab. So, we instead conducted this behavior assay [Fig. R3 (new Fig. S13)] and missed the cellular evidences of D1 or D2 neurons for vmPFC-FrA projections. After all, our present results still provide the important evidence that neural projections from the vmPFC to the FrA, at least the $GABA^+$ ones, can indeed regulate the cocaine-induced locomotor sensitization.

Fig. R2 (new Fig. 5h-k) Both vmPFC-D1 and -D2 neurons project to the FrA with excitatory and inhibitory innervations. (a) Schematic showing the recording of light-evoked EPSCs and

IPSCs in layers II/III of acute FrA slices. Chr2 is expressed through stereotaxic injection of AAV-Flex-ChR2-EYFP into the vmPFC of D1-Cre or D2-Cre mice. **(b, c)** Representative traces and statistics of light-evoked PSCs ($n = 14, 14, 11, 12$) in the FrA. Scale bars, 50 ms, 100 pA. Two-way ANOVA followed by Tukey's multiple comparisons test: D1/D2 effect, $F(1, 47) = 0.69$, $p = 0.41$; EPSC/IPSC effect, $F(1, 47) = 0.006$, $p = 0.94$; multiple comparison tests were all n.s. with $p > 0.9$. **(d)** Percentages of FrA neurons with IPSCs, EPSCs, or both. D1, $n = 60$ cells from 6 mice (33 PSC⁺ cells, 27 PSC⁻ cells); D2, $n = 75$ cells from 7 mice (25 PSC⁺ cells, 50 PSC⁻ cells). Only 20 of the 33 PSC⁺ cells and 14 of the 25 PSC⁺ cells were tested for both EPSCs and IPSCs.

Fig. R3 (new **Fig. S13**) **Inhibitory vmPFC-FrA projections mediate cocaine-induced locomotor sensitization.** Left, schematic showing the virus injection and chemogenetic manipulation of FrA-projecting vmPFC excitatory or inhibitory neurons. Retrograde virus expressing Cre in FrA-projecting vmPFC excitatory or inhibitory neurons was injected bilaterally into the FrA and Cre-dependent Dio-virus expressing mCherry or hM3Dq or hM4Di was injected bilaterally into the vmPFC. Right, statistics of cocaine-induced locomotor sensitization. The behavior paradigm is the same with new Fig. 6. In-group comparisons (between different training days) were analyzed with one-way ANOVA followed by Dunnett's multiple comparisons: CaMKIIα+mCherry, $F(1.116, 6.696) = 10.41$, $p = 0.014$; CaMKIIα+hM3Dq, $F(1.803, 12.62) = 29.05$, $p < 0.001$; CaMKIIα+hM4Di, $F(1.76, 12.32) = 20.96$, $p < 0.001$; vGAT+mCherry, $F(1.734, 12.14) = 27.06$, $p < 0.001$; vGAT+hM3Dq, $F(1.313, 9.194) = 0.09567$, $p = 0.8289$; vGAT+hM4Di, $F(1.782, 12.47) = 32.12$, $p < 0.001$; $n = 8, 8, 8, 8, 7, 8$; multiple comparisons were performed for each group between day 2 or day 3 and day 1. Comparisons between vGAT+hM3Dq and other 5 groups on day 3 were analyzed with one-way ANOVA followed by Dunnett's multiple comparisons: $F(5, 41) = 5.327$, $p < 0.001$. Data are presented as mean \pm s.e.m.. n.s., not significant; * $p < 0.05$; ** $p < 0.01$; *** $p < 0.001$.

3) Additionally, using an elegant approach, the authors demonstrate that vmPFC D1R is an important component that innervates with VTADA inputs and responsible for the cocaine-induced FrA neuronal hypoactivity. However, whether the VTADA-vmPFC D1R circuit is the key element that contributes to cocaine-induced locomotor sensitization behavior may need to be proved.

Response: Following your advice, we combined the micro-infusion of D1R or D2R blocker into the vmPFC with the behavioral tests. We strikingly found that both blockers showed a potent inhibition in the locomotor sensitization [**Fig. R4** (new **Fig. 6e**)], suggesting the essential role of

VTA-DA inputs to the vmPFC in the regulation of cocaine-induced locomotor sensitization.

Fig. R4 (new Fig. 6e) Both vmPFC-D1R and -D2R contribute to the cocaine-induced locomotor sensitization. (a) CPP scores (a1, shown as preference during pre- and post-test) and locomotor distance for sensitization measurement (a2) under bilaterally microinfusion of D1R or D2R antagonist into the vmPFC following i.p. cocaine. CPP scores, $n = 8$ per group; paired t -test between pre- and post-test and unpaired t -test between posttests. Locomotor sensitization, $n = 7, 7, 8$; in-group comparisons (between different training days) were analyzed with one-way ANOVA followed by Dunnett's multiple comparisons: sal, $F(1.749, 10.5) = 17.66$, $p < 0.001$; SCH, $F(1.469, 8.812) = 15.08$, $p = 0.002$; sulph, $F(1.627, 11.39) = 4.407$, $p = 0.04$; multiple comparisons were performed for each group between day 2 or day 3 and day 1. Unpaired t -test for comparisons between SCH/sulp and sal on day 3. Data are presented as the mean \pm s.e.m.. * $p < 0.05$; ** $p < 0.01$; *** $p < 0.001$.

4) To prove that it was the VTA dopaminergic neurons, but not the SNc neurons that play essential roles in mediating cocaine-induced hypoactivity in the FrA neurons, the authors examined FrA activity after a depletion of VTA and SNc-CPu dopaminergic neurons by using 6-OHDA and MPTP approaches, respectively (Line 203-208). Since the MPTP treatment was a systemic administration, can the MPTP metabolite also be transported by DAT into the dopaminergic cells in VTA? Whether the FrA activity change was a concurrent result of VTA and SNc dopaminergic neuronal depletion? It would be better to apply a same treatment (6-OHDA lesion) to deplete SNc region and record FrA activity, as well as provide both the SNc and CPu TH-staining results (Fig S6).

Response: Following your advice, (1) we also used 6-OHDA to deplete dopaminergic neurons in the SNc [Fig. R5a (new Fig. S7d)] and verified the lesion effect with TH-staining of the SNc and the VTA [Fig. R5b (new Fig. S7e)]. We found that although the 6-OHDA treated mice showed a substantial loss of dopamine neurons in the SNc, they still showed a significant hypoactive response to cocaine administration [Fig. R5c (new Fig. S7f)]. (2) Regarding the MPTP treatment (i.p.), we mainly referred to the paper by Guo, *et al.* (2015)¹. This paper showed a relatively limited loss of dopamine neurons in the VTA (~8.1%) compared with that of ~25.2% in the SNc after a 4-day MPTP treatment. Consistently, the dopamine neuron loss is ~8.8% in the VTA vs ~46.7% in the SNc for the 8-day MPTP treatment (Appendix-1)¹. This relative specificity was also reported by another review paper², which shows a much faster depletion effect in the terminals than the somata region. Thus, together with the 6-OHDA (Fig. 2g-i and new Fig. R5) and MPTP (Fig. S7a-c) data, it should be the VTA but not the SNc dopamine neurons that are

necessary for the acute cocaine effect on the FrA.

Fig. R5 (new **Fig. S7d-f**) **Dopaminergic SNc neurons are not necessary for cocaine-induced hypoactivity of the FrA.** (a, b) Schematic of bilateral 6-OHDA injection into SNc and TH-staining (green) showing the loss of dopaminergic somata and fibers in SNc. Scale bar, 500 μ m. (c) Normalized SSCS frequency of FrA neurons in 6-OHDA-treated mice. Sham, n = 168 neurons from 3 mice; 6-OHDA, n = 151 neurons from 3 mice. Wilcoxon test for paired and Mann-Whitney test for unpaired comparisons. ***p < 0.001.

5) By using the *D1-Cre; Ai9* mice and viral tools, the authors showed that *vmPFC-D1R* neurons were involved in mediating cocaine-induced hypoactivity of the FrA neurons. It seems that *D1* and *D2* receptors are both expressed in pyramidal neurons and GABAergic interneurons in mPFC. Although *D1R* are usually considered exerting an excitatory effect while *D2R* producing an inhibitory effect, recent study has demonstrated that *D2R* signaling in the mPFC is often excitatory and these receptors signaling usually complex in mPFC. It would be interesting to know whether the *D2R* also play a crucial role in mediating the neuronal hypoactivity in FrA?

Response: Thanks for your helpful comments. We also noticed the complexity about dopamine signaling in the mPFC, like that cortical neuronal responses to dopamine transmission are usually diversified. Accordingly, *D1R* and *D2R* are proposed not exclusively excitatory or inhibitory^{3,4,5}. Following your advice, we performed new experiments to evaluate the role of *D2R* in cocaine effect with all the same assays regarding *D1R* [Figs. R2, R4, R6 and R10 (new Figs. 5f-k, 6a-e, and S12c, d)]. These results show that chemogenetic manipulation of *vmPFC-D2R* neurons with excitatory hM3Dq also inhibited the FrA activity as *D1R*-neurons did [Fig. R6a, b (new Fig. 6a, b)]. Similarly, both *D1R* and *D2R* antagonism with micro-infusion changed the cocaine's effect on FrA activity [Fig. R6c, d (new Fig. 6c, d)], although the underlying mechanisms could be different. Thus, these results suggest that *vmPFC-D2R* also play a crucial role in the regulation of FrA activity.

Fig. R6 (new Fig. 6a-d) vmPFC modulates FrA through dopamine D1 and D2 receptors. (a, b) Schematic of two-photon Ca^{2+} imaging and statistics of normalized SSCS frequency in the FrA with virus injection (AAV-Dio-hM3Dq/hM4Di/mCherry) into the vmPFC of D1-Cre or D2-Cre mice. Cocaine (i.p.) was injected 30 min after CNO (i.p.) injection. Data for CNO were recorded during 20-30 min after injection and cocaine were 10-20 min. D1-mCh, $n = 225$ neurons from 3 mice; D1-hM3Dq, $n = 296$ neurons from 4 mice; D1-hM4Di, $n = 333$ neurons from 4 mice; D2-hM3Dq, $n = 245$ neurons from 3 mice; D2-hM4Di, $n = 255$ neurons from 3 mice. Wilcoxon test for paired comparisons. (c, d) Schematic of two-photon Ca^{2+} imaging and statistics of normalized SSCS frequency in the FrA following microinfusion of a D1R agonist (SKF-38393, SKF) or antagonist (SCH-23390, SCH) or a D2R agonist (quinpirole, QP) or antagonist (sulpiride, sulp) into the vmPFC following i.p. cocaine in awake mice. Sal, $n = 191$ neurons from 4 mice; SKF, $n = 212$ neurons from 4 mice; SCH, $n = 205$ neurons from 4 mice; QP, $n = 280$ neurons from 5 mice; sulp, $n = 149$ neurons from 4 mice. Wilcoxon test for paired and Mann-Whitney test for unpaired comparisons. * $p < 0.05$; ** $p < 0.01$; *** $p < 0.001$.

Minor comments:

1) Color code in Fig 1 is unclear. Information for each color code in Fig 1 should be provided.

Response: Following your advice, we added the color code for Fig. 1n, which is the same with Fig. 1m as indicated under the x-axis. The other panels were also identified by notes for each group's symbol.

2) The TH-positive signal in FrA of sham group that provide in Fig.S7 was weak, it was difficult to tell the difference between sham and 6-OHDA group. Those figures should be replaced with better quality ones.

Response: Thanks for your advice. We compared the file downloaded from the manuscript submission system to the original one submitted by us. We found that the file was down-sized and thus the thin and sparse TH⁺ fibers in the FrA looks less clear in this figure (original **Fig. S7a**, now as **Fig. S8a**). Following your advice, we used the original pictures and included an enlarged inset to make the anti-TH signal clearer [**Fig. R8a** (new **Fig. S8a**)].

2) *In Fig 4d, it is confusing why the authors illustrated the placement of stimulation electrode was in MFB, not in the VTA. In addition, some of the schematic illustrations appeared in figure legends were “VTA” and some were presented as “DAN”. Clearer information should be provided about the full name of “DAN” when it appears first time in the manuscript.*

Response: Thanks for your question and advice. (1) According to our own^{6,7} and the others' work^{8,9}, it's well-established to get stable evoked terminal signals by stimulating the projection bundles in the MFB which were confined to a limited region, but not the somata region in the VTA which are usually more dispersed. This information has been included in the Methods in the revised manuscript (line 566-569). (2) Following your advice, we changed all “DAN” (dopamine neuron) into “VTA” and revised the figures. Sorry for the confusion.

3) *The pulse number used in Fig.4e “80 p, 80 Hz” seems a very strong stimulation. More explanation would be helpful about how it was selected.*

Response: Thanks for your advice. This protocol was selected based on our previous experience^{6,7} and other papers using phasic stimulation (15-100 Hz) to acquire burst firing of dopamine neurons^{3,8,9,10,11,12}. With CFE *in vivo* recording, it's hard for weak stimulations (< 20 Hz) to elicit a robust dopamine release signal even in the striatum where dopamine release is large. *In vivo* dopamine recordings in other projection regions especially the cortical regions are more difficult and much smaller in amplitude. Thus, we used a stronger but still physiological-achievable stimulation to get the stable signal.

4) *Detail information should be provided about DAT-KO and DAT-CI, SSCS frequency, statistic analyses.*

Response: Following your advice, we revised the manuscript and provided some details for DAT-KO and DAT-CI mice as “DAT-KO mice were homozygous offspring of heterozygous DAT-Cre mice¹³, which was generated by a 5'-UTR knock-in strategy to express Cre recombinase under the control of DAT promoter (line 386-387)” and “DAT-CI mice were kindly gifted by Dr. Howard H. Gu (Ohio State University, USA)¹⁴, which is a knock-in mouse line expressing functional DATs with triple mutations that are insensitive to cocaine (line 388-390)”. We have also provided the references generating DAT-Cre and DAT-CI mouse lines.

Regarding SSCS frequency, we revised the manuscript to provide a definition as “To assess the population activity of stochastic neuronal firing, the averaged frequency of SSCS events collected from all visualized neurons in 6.45 min were quantified as SSCS frequency (events/min/neuron) during the basal (before drug injection) or the drug-treated state (10–20 min after i.p. injection) (line 436-440)”.

In the statistical analyses, we provided details for group analyzing method in line 670-672, biological repeats in line 674-675 and plot types of different data in line 664-665. Thanks for your helpful advices.

Reviewer #2 (Remarks to the Author):

Wang and coworkers provide intriguing new data implicating the frontal associative cortex in cocaine-induced locomotor sensitization. The authors use a variety of powerful techniques to show decreased FrA neuronal activity after cocaine treatment, implicate these neurons in sensitization, and assess the role of dopaminergic projections to vmPFC in the FrA actions. In general the experimental design is strong and the conclusions are well supported by the data. The use of both a calcium dye and GCaMP in the imaging experiments is a nice feature of the study, as the dye allows for better temporal resolution while GCaMP allows for cell-type specific analysis. The lack of effect on CPP is interesting, as it suggests a proscribed role for the FrA. There are aspects of experimental design and analysis that could be improved and/or require additional information. Overall, the findings are new, convincing, and have relevance for understanding the undesirable effects of cocaine.

Response: Thanks for your encouragement and the positive comments. Following all of your and comments and advices below, we conducted new experiments (**Figs. R2, R4, R6, R7, R9, R10**), improved the analysis (**Figs. R7, R8, R11, R12**) and revised the manuscript. For **Figs. R2, R6 and R10**, only the data about D2 receptor and **Fig. R6b** are new. For **Fig. R7**, experiment was repeated to increase the group sizes. For **Fig. R9**, only the data of mCherry group are new. All these new data and analysis have been included in the revised manuscript.

Major Comments:

1. The analysis of the sensitization data in the DREADD experiment does not appear to include a comparison across groups, only a trial-dependent difference in the control group and no such change in the DREADD group. To conclude that the two groups differ it is necessary to perform a between-groups comparison, preferably with a mixed-model ANOVA of some sort. The graph showing these data, figure 1n, also needs labels to designate the different groups.

Response: Following your advice, (1) we performed the between-groups comparison and found the $p = 0.17$ (day 3 between 2 cocaine groups, $n = 8$ per group, ordinary two-way ANOVA followed by Bonferroni's multiple comparisons test: drug effect, $F(1, 28) = 94.14$, $p < 0.001$; virus effect, $F(1, 28) = 2.647$, $p = 0.11$; interaction, $F(1, 28) = 2.664$, $p = 0.11$). To further confirm the phenotype, we repeated the experiment and increased the n values to 11 or 12. The updated results show a significant difference between hM3Dq-coc and mCherry-coc groups on day 3 ($p = 0.0080$: drug effect, $F(1, 43) = 177.4$, $p < 0.001$; virus effect, $F(1, 43) = 5.584$, $p = 0.023$; interaction, $F(1, 43) = 5.954$, $p = 0.019$) but not day 1 ($p > 0.99$: drug effect, $F(1, 43) = 61.52$, $p < 0.001$; virus effect, $F(1, 43) = 0.6946$, $p = 0.41$; interaction, $F(1, 43) = 0.3221$, $p = 0.57$) [**Fig. R7b** (new **Fig.**

1n)]. (2) In the revised manuscript, we labeled the different groups in new Fig. 1n (Fig. R7b) to improve the clarity, and these labels were the same with Fig. 1m (Fig. R7a). Thanks a lot for your advices.

Fig. R7 (new Fig. 1m, n) Acute cocaine-induced hypoactivity of the FrA is necessary for locomotor sensitization. (a) Group data (mean \pm s.e.m.) for CPP score. $n = 11$ (AAV-mCherry, sal), 12, 12, 12. Ordinary two-way ANOVA followed by Bonferroni's multiple comparisons test: drug effect, $F(1, 43) = 49.08$, $p < 0.001$; virus effect, $F(1, 43) = 0.5337$, $p = 0.47$; interaction, $F(1, 43) = 1.364$, $p = 0.25$. (b) Statistics of distance traveled for locomotor sensitization measurement. $n = 11$ (AAV-mCherry, sal), 12, 12, 12. Data are presented as the mean \pm s.e.m.. In-group comparisons (between different training days) were analyzed with one-way ANOVA followed by Dunnett's multiple comparisons: AAV-mCherry (coc), $F(1.921, 21.13) = 30.5$, $p < 0.001$; AAV-hM3Dq (coc), $F(1.406, 15.47) = 0.4257$, $p = 0.59$; multiple comparisons were performed for each cocaine group between day 2 or day 3 and day 1. For comparisons between groups on the same days, ordinary two-way ANOVA followed by Bonferroni's multiple comparisons test: day 1: drug effect, $F(1, 43) = 61.52$, $p < 0.001$; virus effect, $F(1, 43) = 0.6946$, $p = 0.41$; interaction, $F(1, 43) = 0.3221$, $p = 0.57$; day 3: drug effect, $F(1, 43) = 177.4$, $p < 0.001$; virus effect, $F(1, 43) = 5.584$, $p = 0.023$; interaction, $F(1, 43) = 5.954$, $p = 0.019$. n.s., not significant; ** $p < 0.01$; *** $p < 0.001$.

2. The authors should exercise a bit more caution in discussing locomotor sensitization as an indicator of substance use disorders. While locomotor sensitization may be related to incentive sensitization or incubation of craving, the link between these behaviors is not always clear. Discussing future studies using other behavioral measures would be useful in this context.

Response: Following your advice, we read papers and revised our phrasing as "These findings not only suggest the close association of the FrA with locomotor sensitization, a measurement associated with drug-induced plasticity and maybe incentive motivation^{15, 16, 17}, but also define this cortical region as a therapeutic target to specifically reverse cocaine sensitization." (line 320-323) We also emphasized the necessity of studying FrA functions using other behavioral paradigms like self-administration in abstinence, compulsive drug-seeking, and relapse in the

Discussion part (line 324-326) to help outlining the future directions. Thanks for the advices.

3. The authors need to quantify the loss of dopaminergic fibers in the FrA and vmPFC following local 6-OHDA lesions. The main concern is that the lesions in FrA may not be of sufficient magnitude to prevent signaling important for sensitization.

Response: Following your advice, we performed the quantification through the comparison of TH⁺ pixels between sham and 6-OHDA groups. The results [**Fig. R8a, b** (new **Fig. S8a, b**)] show that the averaged pixel number of TH-positive signals for 6-OHDA group is only ~22.2% of the sham group. On the other hand, 6-OHDA depleted DA terminals in the vmPFC to ~13.4% of the sham group [**Fig. R8c** (new **Fig. S11b**)], which should be large enough to make a significant change to the local DA signaling.

Fig. R8 (new **Fig. S8a, b** and **Fig. S11b**) **Depletion of dopaminergic terminals in the FrA or vmPFC by 6-OHDA.** (a) Representative micrographs (same as Fig. 3a) and (b) quantification showing the depletion of dopaminergic terminals in the FrA with TH staining. Scale bar, 50 μ m; pixel size, 0.208 μ m; 1269*504 pixels. Boxed inset in (b) showing the comparison of pixel numbers (TH intensity >120 A.U.) between sham and 6-OHDA groups. n = 7 slices from 5 mice per group. (c) Quantification showing the depletion of dopaminergic terminals in the vmPFC with TH staining. Boxed inset in (c) showing the comparison of pixel numbers (TH intensity >120 A.U.) between sham and 6-OHDA groups. n = 7 slices from 5 mice per group. Data are presented as the mean \pm s.e.m.. Unpaired *t*-test.

4. The experiments showing the efficacy of hM3Dq in altering FrA activity needs a group with no

DREADD expression to ensure that *CNO* has no effects on physiology in the absence of the *DREADD*.

Response: Following your advice, we added a group of mice with no *DREADD* but only *mCherry* expressed in the FrA and found that cocaine could still induce similar hypoactivity [Fig. R9 (new Fig. S3c)]. In addition, the *CNO* injection (0.5 mg/kg) *per se* could not induce significant state change of the FrA neurons as shown in another experiment expressing *mCherry* in *vmPFC-D1* neurons [Fig. R6b (new Fig. 6b)]. Together with the behavioral results (see also Fig. 1n & Fig. S5c), the *hM3Dq*-mediated removing of cocaine effect, but not *CNO* itself, should be the real reason for the lack of locomotor sensitization.

Fig. R9 (new Fig. S3c) Verification of chemogenetic activation of FrA neurons. Normalized SSCS frequency of FrA excitatory neurons 10–20 min after i.p. cocaine (10 mg/kg). Cocaine was injected 30 min after *CNO* (0.5 mg/kg, i.p.) or saline injection. FrA was injected with *AAV-CaMKII α -hM3Dq-mCherry* or *AAV-CaMKII α -mCherry*. *hM3Dq*: Sal, n = 258 neurons from 4 mice; *CNO*, n = 228 neurons from 4 mice. *mCherry*: n = 232 neurons from 4 mice. Wilcoxon test for paired and Mann-Whitney test for unpaired comparisons. *p < 0.05, **p < 0.01, ***p < 0.001.

5. The rationale for the focus on *D1*-expressing *vmPFC* neurons is not explained, although it appears to arise from the involvement of dopamine in the FrA responses. However, it is unclear why *vmPFC* neurons expressing other dopamine receptors (e.g. *D2*) were not examined. It is recognized that the authors could not examine every neuronal subpopulation, but some explanation of how the focus on *D1*-expressing neurons was chosen would be helpful.

Response: Following your advice, in addition to the *D1* experiments, we have added the extensive new experiments of the *D2* experiments in the revised manuscript. Interestingly, *vmPFC-D2* neurons also innervate the FrA with both excitatory and inhibitory projections [Fig. R2 (new Fig. 5h-k); Fig. R10 (new Figs. 5f-g and S12c, d)]. Furthermore, calcium imaging and behavioral tests suggest that both *D1R* and *D2R* play essential roles in the cocaine effect [Figs. R4 and R6 (new Fig. 6)].

Fig. R10 (new **Fig. 5f-g** and **Fig. S12c, d**) **Both vmPFC-D1R and -D2R neurons project to the FrA through excitatory and inhibitory innervations.** (a, b) Schematic and representative micrographs showing retrograde virus (AAV-Retro-CaMKII α -EYFP or AAV-Retro-vGAT-EYFP) injection into the FrA of D1R-Cre or D2R-Cre and Ai9-crossed mice. White arrows indicate the retrograde virus-labeled D1R- or D2R-positive (red) glutamatergic or GABAergic neurons in the vmPFC (IL and DP, coronal slices). Scale bar, 50 μ m. (c) Numbers of labeled excitatory and inhibitory neurons in the vmPFC and percentages of co-labeling with D1 or D2 receptors. Cells were counted in the vmPFC (IL and DP) in an area of 500 \times 500 μ m per slice. CaMKII α , n = 13 slices from 2 D1- and 2 D2-mice; vGAT, n = 12 slices from 2 D1- and 2 D2-mice. Data are presented as the mean \pm s.e.m.. Unpaired *t*-test. ****p* < 0.001. (d) Percentage of excitatory and inhibitory neurons among FrA-projecting vmPFC-D1 or -D2 neurons, according to (c) without consideration of D1R & D2R that expressed in other types of neurons. Numbers in parentheses indicate the average numbers of FrA-projecting vmPFC-D1 or -D2 neurons in one slice within the defined area.

6. For the CPP experiment, it is good that the authors recorded locomotor activity during the test, but they should also include total time in each chamber before and after conditioning in addition to the normalized CPP score. These data could be presented in the supplement, especially if there is no effect of the DREADD treatment on this measure.

Response: Following your advice, we presented the visiting time in each chamber during pretest and posttest [**Fig. R11** (new **Fig. S5a**)] for the CPP experiment in Fig. 1m. We found that both the mCherry-coc and hM3Dq-coc groups showed similar significant stay-time increase in cocaine-paired chamber and decrease in -unpaired chamber after conditioning, while the control groups paired with saline in both chambers showed no significant stay-time change. Thus, these results are consistent with the normalized CPP scores that the hM3Dq treatment didn't change the CPP scores.

Fig. R11 (new **Fig. S5a**) **Time in each chamber during pretest and posttest.** The detailed data of Fig. 1m about the total time spent in each chamber during pretest and posttest which were used for calculating CPP scores. $n = 11$ (AAV-mCherry, sal), 12, 12, 12. Paired t -test for comparisons between pretest and posttest (marked in plot) and unpaired t -test for comparisons between groups (n.s. for all comparisons between coc groups).

7. Was 2p excitation used for both dye- and GCaMP-based calcium imaging? If so, is the 920 nm wavelength optimal for GCaMP excitation?

Response: The 920 nm wavelength is optimal for GCaMP6s excitation and the 830 nm is optimal for Cal-520, which were respectively used in our 2P imaging experiments. We have corrected the typo error as “920 nm for GCaMP6s and 830 nm for Cal-520” in the revised manuscript (line 432-433). Thank you for pointing this out.

8. Figure 4e, the stimulus induced transients appear to be longer lasting in the presence of cocaine, and the authors mention decreased uptake when discussing these results. However, the authors do not present the quantification of this measure. Also, the traces shown in this figure do not decay back to baseline levels so it is not clear if measures of transient duration or area under the curve would be accurate. This group has previously provided evidence that dopamine mediates these amperometric responses in rat (e.g. with voltammetry), but is there comparable evidence in mouse?

Response: Thanks for your helpful comments. (1) We are sorry for the mismatch between the results and the figure. Following your advice, we have presented the statistics of half-height duration (HHD) for the measurements of uptake speed [**Fig. R12c** (new **Fig. 4f**)]. (2) Following your advice, we revised the figure to show the traces with completed decay phase [**Fig. R12b** (new **Fig. 4e**)]. (3) We have previously also validated by voltammetry that the *in vivo* amperometric signals in mouse represent dopamine release in the striatum with electric stimulation in the MFB in a previous work (**Appendix-2**)⁶, which has been cited in the revised manuscript.

Fig. R12 (new Fig. 4d-f) Cocaine enhances dopamine overflow in the vmPFC. (a) *In vivo* amperometric CFE (carbon fiber electrode) recording of dopamine (DA) release in the vmPFC in response to electrical stimulation (E-stim, 1 ms, 80 pulses at 80 Hz) in the MFB (medial forebrain bundle)⁶. (b, c) Representative amperometric traces (I_{amp} , scale bars: 2 s, 30 pA) and statistics of vmPFC DA overflow amplitude and half height duration (HHD) in response to electrical stimulation applied before and after intraperitoneal injection of saline or cocaine (10 mg/kg). Saline, $n = 5$; cocaine, $n = 6$. Paired Student's t -test. * $p < 0.05$, ** $p < 0.01$.

Minor Comments:

a. The title of the paper is inaccurate. Cocaine produces locomotor sensitization and the effects on FrA, but these effects are actually mediated by the neural mechanisms described in the paper.

Response: Following your advice, we have revised the title as “Cocaine induces locomotor sensitization through a dopamine-dependent VTA-mPFC-FrA cortico-cortical pathway”. Thanks for your advice.

b. The term addiction is now thought to be pejorative, and most publications are now switching to substance use disorders. The authors might consider making this change as well. In addition, on line 52 the authors imply that rodents can be addicts, but it is difficult to judge if this is the case given that many of the criteria for SUDs require behaviors that can be assessed in humans but not so readily in rodents.

Response: Following your advice, we have changed the inappropriate using of “addiction” into “substance/drug use disorders” or “dependence” throughout the manuscript. We also changed the “addicts” into “abusers/users” for a better terminology. Thank you so much for the advices.

c. Does the FrA have a role in normal locomotion? The data in figure 1n suggests that it probably does not, but it might be worth discussing this topic.

Response: According to the data and our observations, we didn't find direct role for the FrA in locomotion. We have added this point as “...the FrA shows specificity in regulating sensitization without affecting the normal locomotion or rewarding effect.” in the part of discussion (line 319-320). Thanks a lot for this advice.

d. It would be interesting to know how the calcium signals recorded with GCaMP6s compare to those recorded with the calcium-sensitive dye. Were they slower and/or less numerous with

GCaMP? The underlying question is if GCaMP6s can report spiking activity comparable to that observed with the dye.

Response: According to the literatures, compared to the Cal-520 (**Appendix-3**)¹⁸, GCaMP6s (**Appendix-4**)¹⁹ indeed has a similar-level but slower kinetics (both rise time and decay time) and theoretically may not be able to record some small and short-interval signals while Cal-520 can. Based on our observation and quantification, the averaged basal SSCS frequency by Cal-520 for layer II/III FrA neurons was 2.3 ± 0.05 events/neuron/min, and 2.2 ± 0.08 events/neuron/min by GCaMP6s for excitatory layer II/III FrA neurons (see also **Fig. 1f, h**). Cal-520 seems to record more events (~6%, $p = 0.55$, Mann-Whitney test). Importantly, both the sensors are capable to indicate the cocaine-induced activity changes and produce similar results (frequency reduction of 53% by Cal-520 *versus* 51% by GCaMP6s). Hence, both the sensors can be reliably used for the quantification of cocaine effect.

e. What was the temperature during the brain slice recordings? Could the local drug perfusion alter temperature, i.e. by introducing a room temp solution in a heated bath?

Response: We performed the recording in room temperature bath solution and all the perfusion or recording solutions were balanced to the room temperature before recording, so no temperature variations would be produced when we switch the perfusion channels. Thanks for your question.

Reviewer #3 (Remarks to the Author):

Wang et al identify a novel and interesting circuit underlying the capacity of cocaine to promote locomotor sensitization in mice. Using in vivo 2P calcium imaging, they show that glutamate neuron activity in frontal association cortex (FrA) is strongly reduced for tens of minutes after an injection of cocaine. When these cocaine-induced reductions in FrA activity are counteracted by optogenetic stimulation, locomotor sensitization is not evident, as it is in control mice, following two subsequent injections of cocaine. With clever follow-up experiments, the authors show that the cocaine-induced change in FrA activity is dependent on dopamine and D1 receptor signaling in the ventromedial prefrontal cortex (vmPFC), which they show sends direct, albeit mixed, projections to FrA.

Overall, the manuscript is interesting and novel. With modern techniques, the authors provide multiple converging lines of evidence to support their conclusion that cocaine's influence on FrA activity is indirect, mediated by dopamine and D1 receptor signaling in the vmPFC. Although the work is thorough and convincing, there are two considerable limitations that should be addressed prior to publication.

Response: Thanks for your encouragement and advices. Following your advice, we revised the manuscript with more discussion about literatures and performed new experiments (**Figs. R2, R3, R4, R6, R10, and Appendix-5**) to answer your question about the possible mechanisms underlying how dopamine-dependent vmPFC regulates FrA activity.

Major:

1. The introduction should be rewritten to improve general clarity and include more sophisticated use of psychological language. Specifically,

- a) mice should not be described as addicted, and the use of “addicts” should be omitted completely;
- b) undefined language like “cocaine-adaptive behaviours” should be avoided or specified;
- c) locomotor sensitization should not be equated with incentive saliency without a clear description of the term and citations to support the claim.

Response: Thank you so much for these valuable advices in terminology using. Following your advices:

- a) we have changed “addicts” into “abuser/user” and “addiction” into “substance/drug use disorders” or “dependence” throughout the manuscript;
- b) we revised it as “cocaine-induced place preference and compulsive seeking behaviors” in line 54-55 for unspecified situation;
- c) Following your advice, we have changed “incentive saliency” into “sensitization of incentive motivation^{15, 16, 17}” in line 141, “a measurement for incentive saliency” into “a measurement associated with incentive motivation” in line 321-322, and “reverse incentive-sensitization” into “reverse cocaine sensitization” in line 323.

2. The purported mechanism through which increased D1 receptor signaling in vmPFC produces a sustained reduction in FrA activity is unclear. The authors acknowledge this point without really addressing it. What microcircuits might be involved? Can the authors suggest a mechanism, either by more clearly describing the existing literature or by conducting additional experiments? The sizeable literature on how dopamine affects vmPFC activity should be discussed in more detail, as it may provide insight into how dopamine-induced changes in vmPFC activity can diminish FrA activity. It may be helpful to describe how the effects of dopamine on vmPFC activity seem to be dependent on ongoing network activity (as reviewed in Seamans and Yang, 2004, *Progress in Neurobiology*). Alternatively, the authors could provide evidence that vmPFC-to-FrA projections can actually reduce FrA activity. They could test whether chemogenetic activation of neurons in the vmPFC (whether all of them or just the glutamate or GABA subpopulations) is sufficient to reduce FrA activity. These experiments would shed light on the underlying mechanism by which dopamine signaling in the vmPFC reduces FrA activity.

Response: Thanks for your questions and advices. To address your question “how dopamine-induced changes in vmPFC activity can diminish FrA activity”, we have revised the manuscript with more discussion about the possible mechanisms through combining our new experimental results and literatures. Briefly, based on our revised results with additional important data, (1) both vmPFC-D1R and -D2R play essential roles in the regulation of FrA activity and cocaine sensitization; (2) GABA⁺ projections from the vmPFC to the FrA play critical roles in mediating cocaine-induced FrA hypoactivity and locomotor sensitization.

Regarding the effects of dopamine on vmPFC activity, we agree with you that the modulatory effect is complex and seems to be dependent on ongoing network activity^{5, 20}. Following your

advice, we show our unpublished data, which tested the modulatory effects of dopamine on mPFC neural activity (**Fig. R13**, new **Appendix-5**). Consistently, local application of dopamine (DA, 50 μ M) onto mPFC slices resulted in decreased (89%), no changed (5.6%), or increased (5.6%) firing rate of cortical neurons (**Fig. R13**, new **Appendix-5**), indicating a mixed effect of dopamine (with inhibitory effect dominated) in the mPFC. However, the responses of mPFC neurons to VTA stimulations in the awake mice are more manifold and couldn't be simply quantified as increase or decrease according to recent *in vivo* studies^{3,21}. Mechanistically, although D1 and D2 dopamine receptors are thought to be excitatory and inhibitory, respectively^{5, 20, 22, 23, 24}, there are also opposite results in prefrontal cortex^{25,26}, which show that activating D1R could be inhibitory to the firing of a subset of pyramidal neurons, while D2R could be excitatory. Some earlier publications also report inconsistent results about which receptor(s) underlies DA suppression of prefrontal cortex^{27, 28, 29, 30, 31}, and this suppression may depend on the firing pattern of dopaminergic terminals, inhibitory interneurons, extracellular dopamine concentration and D1R/D2R affinity^{4,5}.

Fig. R13 (new **Appendix-5**) **Dopamine modulates activity of vmPFC neurons.** (a, b) Representative trace and statistics of spontaneous APs in mPFC cortical neurons in response to dopamine (50 μ M) application. (c) Proportion of mPFC neurons with the decreased, no changed, or increased neural activity in response to dopamine application.

Following your advice on adding experiments to test which subpopulation of mPFC neurons is responsible for cocaine-induced hypoactivity of FrA neurons, we performed new experiments to assess roles of D2R neurons in cocaine effect. The anatomical results show that vmPFC-D2 neurons also project to the FrA with both excitatory and inhibitory projections [**Fig. R2** (new **Fig. 5h-k**); **Fig. R10** (new **Fig. 5f-g** and **S12c, d**)]. Excitatory manipulation of vmPFC-D2 neurons by hM3Dq also inhibited the FrA as D1-neurons did [**Fig. R6b** (new **Fig. 6b**)], although with different extent and maybe different mechanisms. Furthermore, both D1R and D2R antagonisms by micro-infusion changed the cocaine effect on FrA activity [**Fig. R6d** (new **Fig. 6d**)] and blocked the cocaine-induced sensitization [**Fig. R4** (new **Fig. 6e**)]. Thus, these results suggest that both vmPFC-D1R and -D2R play essential roles in the regulation of FrA activity and cocaine sensitization. Together, these new data (including new Figs. **5f-k**, **6a-e** and **S12c-d**) have been included in the revision.

To exam which vmPFC-FrA projection(s) contributes to the behavior of cocaine sensitization, we combined retrograde virus (Retro-CaMKII α /vGAT-Cre in the FrA) and Cre-dependent virus (Dio-DREADD in the vmPFC) to manipulate FrA-projecting vmPFC neurons (Glu⁺ or GABA⁺) with hM3Dq or hM4Di. Strikingly, only GABA⁺ FrA-projecting vmPFC neurons expressing

excitatory hM3Dq profoundly abolished the locomotor sensitization compared to the other 5 groups [**Fig. R3** (new **Fig. S13**)]. Together, these results provide meaningful evidence that projections from the vmPFC to the FrA (activating the GABA⁺ neurons) may regulate the FrA activity and locomotor sensitization. In addition, we also raised some open questions about the vmPFC-FrA mechanisms underlying acute cocaine effect on the FrA in the Discussion (line 365-369 in the manuscript).

Minor:

1. *The rationale for recording FrA activity in response to cocaine is weak. It may be possible to bolster the rationale by discussing previous literature that characterizes the effects of dopamine on mPFC activity within the context of substance use disorders.*

Response: Following your advice, we have revised the manuscript by discussing the literatures about dopamine regulation of mPFC activity in the introduction as mentioned above (line 71-82 in the manuscript).

2. *This statement in the introduction, “...it is not clear how and to what extent the PFC of an awake mouse is affected by acute cocaine” is dismissive of prior work in this area. It would be better to describe what actually is and is not known about the acute actions of cocaine in the PFC.*

Response: Following your advice, we have revised the statement to “To address how and to what extent the multiple PFC sub-regions of an awake mouse are affected by acute cocaine ... (line 94-95 in the manuscript)”.

In addition, we found a relevant paper³¹ reporting the acute cocaine-induced mPFC suppression through *in vivo* intracellular recording of mPFC neurons in anesthetized rats (line 72 in the manuscript). We also cited a fMRI study of human brain exposed to acute cocaine involving the PFC³² (line 308 in the manuscript).

We thank you very much for your helpful and valuable comments!

References:

1. Guo L, *et al.* Dynamic rewiring of neural circuits in the motor cortex in mouse models of Parkinson's disease. *Nat Neurosci* **18**, 1299-1309 (2015).
2. Meredith GE, Rademacher DJ. MPTP mouse models of Parkinson's disease: an update. *J Parkinsons Dis* **1**, 19-33 (2011).
3. Lohani S, Martig AK, Deisseroth K, Witten IB, Moghaddam B. Dopamine Modulation of Prefrontal Cortex Activity Is Manifold and Operates at Multiple Temporal and Spatial Scales. *Cell Rep* **27**, 99-114 e116 (2019).
4. Trantham-Davidson H, Neely LC, Lavin A, Seamans JK. Mechanisms underlying differential D1 versus D2 dopamine receptor regulation of inhibition in prefrontal cortex. *J Neurosci* **24**, 10652-10659 (2004).
5. Seamans JK, Yang CR. The principal features and mechanisms of dopamine modulation in the prefrontal cortex. *Prog Neurobiol* **74**, 1-58 (2004).
6. Xu H, *et al.* Striatal dopamine release in a schizophrenia mouse model measured by electrochemical amperometry in vivo. *Analyst* **140**, 3840-3845 (2015).
7. Wang SR, *et al.* Role of vesicle pools in action potential pattern-dependent dopamine overflow in rat striatum in vivo. *J Neurochem* **119**, 342-353 (2011).
8. Ewing AG, Bigelow JC, Wightman RM. Direct in vivo monitoring of dopamine released from two striatal compartments in the rat. *Science* **221**, 169-171 (1983).
9. Kuhr WG, Wightman RM. Real-time measurement of dopamine release in rat brain. *Brain Res* **381**, 168-171 (1986).
10. Schultz W, Dayan P, Montague PR. A neural substrate of prediction and reward. *Science* **275**, 1593-1599 (1997).
11. Hyland BI, Reynolds JN, Hay J, Perk CG, Miller R. Firing modes of midbrain dopamine cells in the freely moving rat. *Neuroscience* **114**, 475-492 (2002).
12. Kiyatkin EA, Rebec GV. Heterogeneity of ventral tegmental area neurons: single-unit recording and iontophoresis in awake, unrestrained rats. *Neuroscience* **85**, 1285-1309 (1998).
13. Zhuang X, Masson J, Gingrich JA, Rayport S, Hen R. Targeted gene expression in dopamine and serotonin neurons of the mouse brain. *J Neurosci Methods* **143**, 27-32 (2005).
14. Chen R, *et al.* Abolished cocaine reward in mice with a cocaine-insensitive dopamine transporter. *Proc Natl Acad Sci U S A* **103**, 9333-9338 (2006).
15. Berridge KC, Robinson TE. Liking, wanting, and the incentive-sensitization theory of addiction. *Am Psychol* **71**, 670-679 (2016).
16. Bocklisch C, *et al.* Cocaine disinhibits dopamine neurons by potentiation of GABA transmission in the ventral tegmental area. *Science* **341**, 1521-1525 (2013).
17. Steketee JD, Kalivas PW. Drug wanting: behavioral sensitization and relapse to drug-seeking behavior. *Pharmacol Rev* **63**, 348-365 (2011).
18. Tada M, Takeuchi A, Hashizume M, Kitamura K, Kano M. A highly sensitive fluorescent indicator dye for calcium imaging of neural activity in vitro and in vivo. *Eur J Neurosci* **39**, 1720-1728 (2014).
19. Chen TW, *et al.* Ultrasensitive fluorescent proteins for imaging neuronal activity. *Nature*

- 499**, 295-300 (2013).
20. Tritsch NX, Sabatini BL. Dopaminergic modulation of synaptic transmission in cortex and striatum. *Neuron* **76**, 33-50 (2012).
 21. Decot HK, *et al.* Coordination of Brain-Wide Activity Dynamics by Dopaminergic Neurons. *Neuropsychopharmacology* **42**, 615-627 (2017).
 22. Lahiri AK, Bevan MD. Dopaminergic Transmission Rapidly and Persistently Enhances Excitability of D1 Receptor-Expressing Striatal Projection Neurons. *Neuron* **106**, 277-290 e276 (2020).
 23. Gullledge AT, Jaffe DB. Dopamine decreases the excitability of layer V pyramidal cells in the rat prefrontal cortex. *J Neurosci* **18**, 9139-9151 (1998).
 24. Luo Z, Volkow ND, Heintz N, Pan Y, Du C. Acute cocaine induces fast activation of D1 receptor and progressive deactivation of D2 receptor striatal neurons: in vivo optical microprobe [Ca²⁺]_i imaging. *J Neurosci* **31**, 13180-13190 (2011).
 25. Lancon K, Qu C, Navratilova E, Porreca F, Seguela P. Decreased dopaminergic inhibition of pyramidal neurons in anterior cingulate cortex maintains chronic neuropathic pain. *Cell Rep* **37**, 109933 (2021).
 26. Wang Y, Goldman-Rakic PS. D2 receptor regulation of synaptic burst firing in prefrontal cortical pyramidal neurons. *Proc Natl Acad Sci U S A* **101**, 5093-5098 (2004).
 27. Gorelova N, Seamans JK, Yang CR. Mechanisms of dopamine activation of fast-spiking interneurons that exert inhibition in rat prefrontal cortex. *J Neurophysiol* **88**, 3150-3166 (2002).
 28. Parfitt KD, Gratton A, Bickford-Wimer PC. Electrophysiological effects of selective D1 and D2 dopamine receptor agonists in the medial prefrontal cortex of young and aged Fischer 344 rats. *J Pharmacol Exp Ther* **254**, 539-545 (1990).
 29. Sesack SR, Bunney BS. Pharmacological characterization of the receptor mediating electrophysiological responses to dopamine in the rat medial prefrontal cortex: a microiontophoretic study. *J Pharmacol Exp Ther* **248**, 1323-1333 (1989).
 30. Yang CR, Seamans JK. Dopamine D1 receptor actions in layers V-VI rat prefrontal cortex neurons in vitro: modulation of dendritic-somatic signal integration. *J Neurosci* **16**, 1922-1935 (1996).
 31. Trantham-Davidson H, Lavin A. Acute cocaine administration depresses cortical activity. *Neuropsychopharmacology* **29**, 2046-2051 (2004).
 32. Kufahl PR, *et al.* Neural responses to acute cocaine administration in the human brain detected by fMRI. *Neuroimage* **28**, 904-914 (2005).

Appendix-1: MPTP-induced loss of dopaminergic neurons in SNc and VTA (Guo, *et al.* 2015)¹

Supplementary Figure 1

Confocal images of sections processed for TH immunoreactivity.

Sections taken from control (a-c), 1 day MPTP (d-f), 4 days MPTP (g-i), and 8 days MPTP (j-l) treated mice. Left (a, d, g, j): low magnification. White box indicates the region of interest (ROI). Right: high magnification images of TH+ dopaminergic neurons inside the ROI. Following MPTP-treatment, there is significant loss of dopamine neurons in VTA (b, e, h, k) and SNc (c, f, i, l). Scale bar = 100 μ m for panels j and 50 μ m in panels b, c, e, f, h, i, k and l. The loss of dopamine neurons in SNc after 1d MPTP-treatment: $5.2 \pm 8.3\%$, $n = 4$ mice, $P = 0.3429$, Mann-Whitney; after 4d MPTP-treatment: $25.2 \pm 2.8\%$, $n = 4$ mice, $P = 0.0286$, Mann-Whitney; after 8d MPTP-treatment: $46.7 \pm 2.9\%$, $n = 4$ mice, $P = 0.0286$, Mann-Whitney. The loss of dopamine neurons in VTA after 1d MPTP-treatment: $2.8 \pm 3.0\%$, $n = 4$ mice, $P = 0.8557$, Mann-Whitney; after 4d MPTP-treatment: $8.1 \pm 1.6\%$, $n = 4$ mice, $P = 0.0408$, Mann-Whitney; after 8d MPTP-treatment: $8.8 \pm 1.1\%$, $n = 4$ mice, $P = 0.0294$, Mann-Whitney.

Appendix-2: Following electric stimulation in the MFB, the amperometric signal (I_{amp}) represents dopamine release in mouse striatum *in vivo* (Xu *et al.* 2015) ⁶.

Fig. 1 Methodology of electric-chemical amperometry recording of DA overflows in anesthetic mice striatum *in vivo*. (A) Circuit diagram for the *in vivo* amperometric recording system. (B) Diagram of electrode positions in DA overflow recorded in the mouse striatum *in vivo*. (C) An example trace for DA overflow (I_{amp}) following 36 stimulus pulses at 80 Hz applied to the MFB. Amplitude, HHD and V_{max} are defined as illustrated. (D) DA signals detected by fast scan cyclic voltammetry (FSCV) *in vivo*. Left panel, time course of a DA overflow signal ($I_{FSCV}-t$ curve) derived from FSCV recordings at 620 mV. Inset, representative FSCV voltammograms of the DA signal in the striatum *in vivo* (black line) and 5 μ M DA solution *in vitro* (dashed gray line). Right panel, a corresponding $I_{FSCV}-V-t$ pseudocolour plot,¹⁹ to the left, I_{FSCV} .

Appendix-3: Properties of Cal-520 (Tada, *et al.* 2014)¹⁸

FIG. 1. Properties of Cal-520 examined in layer 2/3 pyramidal cells in the barrel cortex in acute brain slices. (A) Representative image of whole-cell patch-clamp recording from a pyramidal cell in the barrel cortex. Dashed lines indicate the patch pipette. A magnified image is shown in the right panel. The red line indicates the position of the linescan. (B) Representative traces of fluorescence response ($\Delta F/F$) of Cal-520 and OGB-1 in response to one, two, four, and eight spike trains at 20 Hz. Mean amplitude (C) and SNR (D) of Cal-520 and OGB-1 in response to one to eight spike trains at 20 Hz. Error bars indicate SEM ($n = 7$ cells each, $*P < 0.05$, $**P < 0.01$). (E) Rise times and decay time constants of calcium transients induced by single APs. Error bars indicate SEM ($n = 7$ cells each). (F) Trial-averaged fluorescence transients of Cal-520 to five spike trains at different frequencies. Bottom: responses to 10 and 20 Hz spike trains are shown in an expanded time scale. Gray lines indicate individual sweeps ($n = 10$). Red thick lines indicate average traces. Stimuli are indicated by the black vertical lines.

Appendix-4: Properties of GCaMP6s (Chen, *et al.* 2013)¹⁹

Figure 1 | GCaMP mutagenesis and screening in dissociated neurons. **a**, GCaMP structure^{27,28} and mutations in different GCaMP variants relative to GCaMP5G. **b**, Responses averaged across multiple neurons and wells for GCaMP3, 5G, 6f, 6m, 6s, and OGB1-AM. Top, fluorescence changes in response to 1 action potential. Bottom, 10 action potentials. **c**, Screening results, 447 GCaMPs. Top, fluorescence change in response to 1 action potential (vertical bars, $\Delta F/F_0$; green bar, OGB1-AM, left; black bars, single GCaMP mutations; red bars, combinatorial mutations; blue, GCaMP6 indicators) and significance values for different action potential stimuli (colour plot). Middle, half decay time after 10 action potentials. Bottom, resting fluorescence, F_0 normalized to nuclear mCherry fluorescence. Red line, GCaMP3 level; green line, GCaMP5G level; blue line, OGB1-AM level. AP, action potential. **d-g**, Comparison of GCaMP sensors and OGB1-AM as a function of stimulus strength (colours as in **b**). **d**, Response amplitude. **e**, Signal-to-noise ratio (SNR). **f**, Half decay time. **g**, Time to peak (after stimulus offset). Error bars correspond to s.e.m ($n = 300, 16, 8, 11, 13$ and 11 wells for GCaMP3, GCaMP5G, OGB1-AM, 6f, 6m and 6s, respectively).

Appendix-5: Dopamine modulates activity of vmPFC neurons. (a, b) Representative trace and statistics of spontaneous APs in mPFC cortical neurons in response to dopamine (50 μ M) application. (c) Proportion of mPFC neurons with the decreased, no changed, or increased neural activity in response to dopamine application. (Note, this appendix-5 is part of author Changhe Wang's manuscript currently under review).

REVIEWERS' COMMENTS

Reviewer #1 (Remarks to the Author):

The authors are very responsive to the original comments, and thoroughly addressed the concerns by performing the recommended experiments and including additional information. The new experimental data and more information included in the revision have strengthened the manuscript enormously. No further comments.

Reviewer #2 (Remarks to the Author):

The revisions have greatly improved the manuscript and addressed the many points raised in review. There are only two additional points.

1. It is interesting to see that local infusion of a D2 antagonist blocks sensitization. Does this treatment have any effect on baseline locomotion?
- 2.. The grammar and usage in the manuscript requires additional editing.

Reviewer #3 (Remarks to the Author):

The authors nicely addressed all my comments. I support publication. The work is interesting and novel. With modern techniques, the authors provide multiple converging lines of evidence to support their conclusion that cocaine's influence on FrA activity is mediated by dopamine signaling in the vmPFC.

Point-by-point response to the reviewers' comments

Reviewer #1 (Remarks to the Author):

The authors are very responsive to the original comments, and thoroughly addressed the concerns by performing the recommended experiments and including additional information. The new experimental data and more information included in the revision have strengthened the manuscript enormously. No further comments.

Response: Thanks for your encouragement on this work. Your advices helped a lot to strengthen the revision.

Reviewer #2 (Remarks to the Author):

The revisions have greatly improved the manuscript and addressed the many points raised in review. There are only two additional points.

1. It is interesting to see that local infusion of a D2 antagonist blocks sensitization. Does this treatment have any effect on baseline locomotion?

Response: Following your advice, we have performed new experiments and found that local perfusion of haloperidol in the mPFC showed no effect on baseline locomotion (Fig. R14).

Fig. R14. D2R antagonist treatment in the mPFC showed no effect on baseline locomotion. a, Schematic of bilateral cannula application of the D2R antagonist haloperidol (Halo) in the mPFC. **b,c,** Statistics of traveling distance and maximal speed of mice with or without the local administration of haloperidol in the mPFC.

2. The grammar and usage in the manuscript requires additional editing.

Response: Thanks for your comments. Following your advice, we have revised the manuscript with additional editing.

Reviewer #3 (Remarks to the Author):

The authors nicely addressed all my comments. I support publication. The work is interesting and novel. With modern techniques, the authors provide multiple converging lines of evidence to support their conclusion that cocaine's influence on FrA activity is mediated by dopamine signaling in the vmPFC.

Response: Thanks for your encouragement. Your advices have helped a lot to improve the manuscript.